# BOOSTED TREES ON A DIET: COMPACT MODELS FOR RESOURCE-CONSTRAINED DEVICES

**Nina Herrmann[1]**[*] **Jan Stenkamp[1]**[*] **Benjamin Karic[1], Stefan Oehmcke[2], Fabian Gieseke[1,3]**
[1]University of Münster, [2]University of Rostock, [3]University of Copenhagen
`{nina.herrmann,jan.stenkamp}@uni-muenster.de`

## ABSTRACT

Deploying machine learning models on compute-constrained devices has become a key building block of modern IoT applications. In this work, we present a compression scheme for boosted decision trees, addressing the growing need for lightweight machine learning models. Specifically, we provide techniques for training compact boosted decision tree ensembles that exhibit a reduced memory footprint by rewarding, among other things, the reuse of features and thresholds during training. Our experimental evaluation shows that models achieved the same performance with a compression ratio of 4–16x compared to LightGBM models using an adapted training process and an alternative memory layout. Once deployed, the corresponding IoT devices can operate independently of constant communication or external energy supply, and, thus, autonomously, requiring only minimal computing power and energy. This capability opens the door to a wide range of IoT applications, including remote monitoring, edge analytics, and real-time decision making in isolated or power-limited environments.

## 1 INTRODUCTION

Modern Internet of Things (IoT) techniques have paved the way for new applications in domains such as home automation, healthcare, agriculture, or industry (Khanna & Kaur, 2020). For instance, in the context of home automation, sensors can be used to automate (smart) lighting and (smart) heating. Another example is using sensor data for predictive maintenance to prevent machine failures and enhance productivity. Microcontrollers are a key building block of these IoT applications, often equipped with sensors to measure parameters such as temperature, humidity, pressure, or vibrations. A key characteristic of such devices is that they generally have very limited computing and memory resources (Ojo et al., 2018). For instance, the broadly used open-source *Arduino Uno R4 Minima* board is equipped with a 32-bit *Renesas RA4M1* microcontroller and an *Arm Cortex-M4* processor running at 48 MHz, 32 KB main memory (RAM), 256 KB flash storage, and 1 KB electrically erasable programmable read-only memory (EEPROM). There are also microcontrollers with even less memory and computing resources, such as the *Arduino Nano* board. Furthermore, the available resources must be shared among the operating system, sensing data, data-processing programs, and machine learning models. Another characteristic is that IoT microcontrollers are often designed for energy efficiency. This makes them well-suited for being deployed in remote locations, where a continuous power supply is not available. Under ideal conditions, such devices can run for several months or even years via batteries. Such applications are also supported by specialized communication protocols such as LoRa.[1]

To minimize energy-intensive data transfers and to enable real-time processing, the on-site analysis of data on the IoT device is preferable if possible. Embedding machine learning methods directly into IoT nodes addresses this need. The key challenge in embedded machine learning is to minimize both compute and memory requirements to enable execution on resource-constrained devices while, at the same time, preserving model quality. This has given rise to the concept of *Tiny Machine Learning (TinyML)* models (Gural & Murmann, 2019; Reddi et al., 2021; Kumar et al., 2017; Warden & Situnayake, 2019). By sufficiently reducing resource demands, the "tiny" models can be run

---

[*]Equal contribution.
[1]LoRa enables small volumes of data to be transferred over several kilometres (Mayer et al., 2019).

directly on the microcontrollers, enabling compact "smart" devices. An example of a corresponding IoT application is sketched in Figure 1.

**Contribution:** We propose a framework that allows for compressing boosted decision tree ensembles, one of the most widely used machine learning models for structured data. Our approach, referred to as Trees on a Diet (`ToaD`), relies on (1) regularizers that encourage the reuse of features and thresholds during training and on (2) a specialized memory layout to store the resulting trees. More precisely, we utilize global lookup tables for both features and thresholds, and we store the trees without pointers using an adapted bit-wise encoding. Overall, these modifications lead to tree ensembles with reduced memory requirements, without sacrificing model quality. We showcase the effectiveness of our approach by assessing the quality-memory trade-off in our experimental evaluation. Our results indicate that the adapted training process yields models of comparable performance while achieving compression ratios of 4-16x compared to other baselines.

## 2  BACKGROUND

We begin with the background on boosted decision trees and tiny machine learning models.

### 2.1  BOOSTED DECISION TREES

In contrast to single decision trees, tree ensembles aggregate the outputs of multiple trees. For instance, a random forest (RF) (Breiman, 2001) is obtained by constructing the trees independently from each other, introducing randomness to the construction process so that a set of different trees is obtained. In contrast, boosted decision trees are built in an incremental manner, one tree at a time. Two prominent frameworks in this context are XGBoost (Chen & Guestrin, 2016) and LightGBM (Ke et al., 2017).

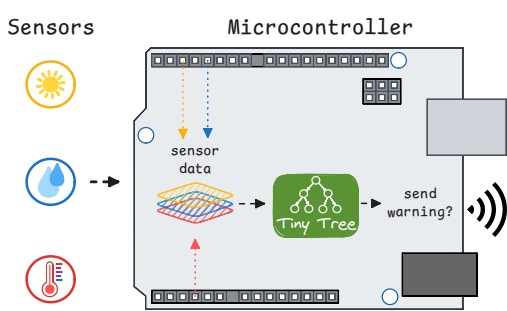

Figure 1: A machine learning model (decision tree) on a microcontroller processes multi-sensor data locally and transmits only relevant events, reducing energy costs; the decision tree must have a minimal compute and memory footprint.

Let $\mathcal{T} = \{(\mathbf{x}_1, y_1), \ldots, (\mathbf{x}_n, y_n)\} \subset \mathbb{R}^d \times \mathcal{Y}$ be a training (multi)set of data points $\mathbf{x}_i \in \mathbb{R}^d$ with associated labels $y_i \in \mathcal{Y}$. Here, $\mathcal{Y}$ is the set of labels. For regression, we have $\mathcal{Y} = \mathbb{R}$, whereas one is given a finite set $\mathcal{Y} = \{p_1, \ldots, p_c\}$ of classes for classification scenarios. Both XGBoost and LightGBM model ensembles are built in an additive way, resulting in a tree ensemble model $T$, whose prediction $T(\mathbf{x})$ for a new data point $\mathbf{x} \in \mathbb{R}^d$ is based on the sum of predictions made by $K$ individual decision trees $t_k$, i.e., $T(\mathbf{x}) = \sum_{k=1}^{K} t_k(\mathbf{x})$, where $t_k(\mathbf{x})$ denotes the prediction made by the tree $t_k$ for an instance $\mathbf{x} \in \mathbb{R}^d$. Such ensembles are built in a way to minimize

$$\sum_{i=1}^{n} \mathcal{L}\left(y_i, T(\mathbf{x}_i)\right) + \sum_{k=1}^{K} \Omega\left(t_k\right), \tag{1}$$

where $\mathcal{L} : \mathcal{Y} \times \mathcal{Y} \to \mathbb{R}^+$ is a suitable loss function and where $\Omega\left(t_k\right)$ specifies the complexity of the tree $t_k$. Constructing an optimal tree ensemble $T$ w.r.t. Equation (1) is generally not feasible. Instead, one typically resorts to $K$ boosting rounds, and in each round, one new tree is added to the ensemble built so far so that the Equation (1) is minimized. More precisely, in boosting round $m \geq 1$, one considers the ensemble $T_{m-1} := \sum_{k=1}^{m-1} t_k$ of the trees built so far, starting with $T_0 := 0$, and aims to find the next best tree $t_m$ that minimizes $\sum_{i=1}^{n} \mathcal{L}\left(y_i, T_{m-1}(\mathbf{x}_i) + t_m(\mathbf{x}_i)\right) + \sum_{k=1}^{m} \Omega(t_k)$. The decision tree $t_m$ itself is also constructed in a greedy manner, starting with the root of the tree being recursively split. To penalize "complex" trees, one usually resorts to $\Omega(t_m) = \gamma L + \frac{1}{2}\lambda \sum_{j=1}^{L} v_j^2$ as regularizer, where $L$ is the number of leaves and $v_1, \ldots, v_L \in \mathbb{R}$ the associated leaf values of $t_m$.

### 2.2  RELATED WORK

Decision trees and decision tree ensembles, such as Gradient Boosted Decision Trees (GBDT), have remained very popular, despite the rise of deep learning models (Grinsztajn et al., 2022), especially

for structured (tabular-like) data. They are also generally easy to interpret and need few computational resources. Recent advances in tree-based models have also introduced several approaches to enhance their efficiency. For instance, several works focused on improving the training or inference speed of GBDT by means of array structures (Lucchese et al., 2017; Ye et al., 2018) or via the quantization of gradient statistics (Shi et al., 2022; Jiang et al., 2018; Devos et al., 2020). Quantization of tree parameters was employed by Koschel et al. (2023), who adapted variants of QuickScorer (Lucchese et al., 2017) to enable deployment on IoT devices. Moreover, quantization of GBDT for resource-constrained devices was investigated by Alsharari et al. (2025) and Wang et al. (2023), both targeting FPGAs. Reduction of model size and latency is the motivation for different post-training pruning techniques (Liu & Mazumder, 2023; Guo et al., 2018), some including the refinement of leave values (Devos et al., 2025; Emine et al., 2025; Buschjäger & Morik, 2023). Increased efficiency of tree models by design or within the training process is addressed by works from Kumar et al. (2017) for single decision tree sizes, Ponomareva et al. (2017) for multiclass classification, and Peter et al. (2017) for efficient evaluation of deep trees considering feature acquisition and tree evaluation costs. Further work focusing primarily on the optimization of RFs include Ren et al. (2015), Alkhoury et al. (2025), and Nan et al. (2016). Due to a lack of space, we refer to Appendix C for a more detailed discussion of related approaches.

In this work, we focus on gradient-boosted trees. For subsequent deployments, our aim is to minimize the memory footprint of the ensemble during the training process, while simultaneously ensuring a good model fit and compact model size. We extend the work of Peter et al. (2017) by defining suitable feature costs that aim at reducing the bits required to encode feature indices. We also introduce a corresponding cost regularizer for split thresholds and leaf values, and provide a specialized encoding for the induced binary trees. Moreover, our memory layout includes global threshold arrays shared by all learners.

## 3 APPROACH

Pruning or quantization techniques are typically applied before or after the training to reduce the size of trees (see Section 4). However, such methods generally cannot exploit task-specific compression potential. For instance, they are not designed to incorporate the potential in memory saving of feature sharing or the reuse of (leaf/split) thresholds within a single tree or across all trees in the ensemble. Our framework exploits this potential by penalizing unused features and thresholds when growing new trees. In combination with a corresponding memory layout, this yields a substantially smaller memory footprint with reused features and thresholds being stored more compactly.[2]

### 3.1 TRAINING COMPRESSED TREES

As sketched above, boosted tree ensembles are built in an incremental manner, and in each boosting round $m$, a new tree $t_m$ is added to the ensemble. The tree $t_m$ is, in turn, also built in a greedy manner by iteratively splitting leaves (starting with the root) if this leads to a better objective (if not, the construction process stops). To assess the quality of such a leaf split, an associated *gain* $\Delta$ is computed (Chen & Guestrin, 2016). More specifically, for a leaf associated with a set $I$ of training indices, a split along feature dimension $i \in \{1, \ldots, d\}$ with respect to threshold $\mu$ induces a potential gain $\Delta(I, i, \mu) \in \mathbb{R}$ (which may also be negative). Leaves are split as long as some split yields a positive gain, always choosing the leaf and split with the highest gain.

The standard gain does not promote the reuse of features or thresholds. Following Peter et al. (2017), we introduce an additional regularizer based on the set of features $F_U \subseteq \{1, \ldots, d\}$ and thresholds $\mathcal{T}^f \subset \mathbb{R}$ with $f \in F_U$ that have already been used by the trees $t_1, \ldots, t_m$ built so far (including the current tree $t_m$). The memory layout detailed below allows for storing those features and thresholds in a much more compact manner. In particular, features and thresholds that have already been

---

[2]Our work targets resource-constrained devices where memory, not latency or energy, is the main bottleneck. In many applications, the model size determines whether a deployment is feasible or not. It is worth pointing out that local on-device inference is generally far more energy-efficient than transmitting data to a remote server and also incurs only minimal latency. Technically, our compression scheme adds only minimal overhead (a few bit-wise operations), so we expect the impact on latency and energy per prediction to be negligible. Preliminary insights on prototypical implementations of `ToaD` and LightGBM deployed on different microcontrollers are discussed in Appendix E.1.

Figure 2: High-level sketch of the memory layout used to store an ensemble of boosted decision trees. The first part stores some metadata, such as the number $K$ of boosted trees or the maximum depth of all trees. The following three parts encode the used features, thresholds, and leaf values. Finally, the references to the features and thresholds for the individual trees are stored.

used in previous trees contribute only marginally to the overall space consumption and are therefore essentially "free of charge". A simple linear regularizer that favors such a reuse of features and thresholds is given by

$$\Omega_l(t_m) = \Omega(t_m) + \iota \cdot |F_U| + \xi \cdot \sum_{f \in F_U} |\mathcal{T}^f|, \tag{2}$$

where $\iota, \xi \in \mathbb{R}^+$ are user-defined hyperparameters. Thus, using a new feature that has not been used so far leads to an increase of $|F_U|$ by 1, and, hence, to an increase of the objective (1) by $\iota$. Accordingly, if a new threshold is used for a feature $f \in F_U$, the objective is increased by $\xi$.[3] Using this modified regularizer leads to the following modified gain

$$\Delta_l(I, i, \mu) = \Delta(I, i, \mu) - s_f \iota - s_t \xi, \tag{3}$$

where $s_f = 1$ in case a new feature (index) is used and $s_f = 0$ otherwise, and $s_t = 1$ in case a new threshold is used and $s_t = 0$ otherwise, see Appendix A for the derivations.

## 3.2 MEMORY LAYOUT

Our memory layout leverages the reuse of feature indices and thresholds enforced by the modified regularizer. A high-level sketch of the overall memory layout is shown in Figure 2. In a nutshell, it reduces the memory footprint of boosted tree ensembles through two mechanisms:

1. *Bit-wise encoding:* Encoding the information in a bit-wise manner allows to store a minimal representation of information compared to the use of higher level data types that may use non-minimal representations (e.g. a `bool` occupies eight bits in memory in `C`).

2. *Shared thresholds and leaf values:* Global arrays are used to store threshold and leaf values, which are referenced within internal tree nodes and leaves. Sharing values across trees can substantially reduce the bits needed for storing thresholds and leaf weights.

The memory layout comprises five components. The first stores metadata, including the number $K$ of trees, the maximum tree depth, the number $|F_U|$ of used features, and the maximum number of thresholds $\max_{f \in F_U} |\mathcal{T}^f|$ associated with any feature. In addition, three global arrays and the individual decision trees $t_1, \ldots, t_K$ are stored. Figure 3 illustrates the bit-wise encoding of a simple model with two exemplary decision trees. We now detail the individual components.

### 3.2.1 BIT-WISE ENCODING

Boosted ensembles typically employ shallow, nearly balanced trees with a small depth (e.g., a depth of up to 5). Such trees can be stored efficiently using pointer-less schemes. More precisely, the root is stored at index $i = 0$, and for a node at index $i$, the left child is stored at index $2 \cdot i + 1$ and the right child at index $2 \cdot i + 2$. For example, in Figure 3, two such array-based representations are given. Here, the root $n_1$ of tree $t_1$ is stored at index 0, and its two children $n_2$ and $n_3$ are stored at indices 1 and 2, respectively. We distinguish between internal nodes and leaf nodes:

---

[3]For example, if a feature corresponds to temperature values, it may be sufficient to restrict the set of admissible thresholds to, e.g., 0 and 20 degrees Celsius. Alternative regularization schemes are also conceivable. Another option is, for instance, $\Omega_e(t_m) = \Omega(t_m) + \iota \cdot \sum_{j=1}^{|F_U|} j + \xi \cdot \sum_{j=1}^{p} j$ with $p = \sum_{f \in F_U} |\mathcal{T}^f|$, which imposes an exponentially increasing penalty on the number of distinct features and thresholds. In practice, however, the linear regularizer has been shown to be highly effective and is, hence, used in this work.

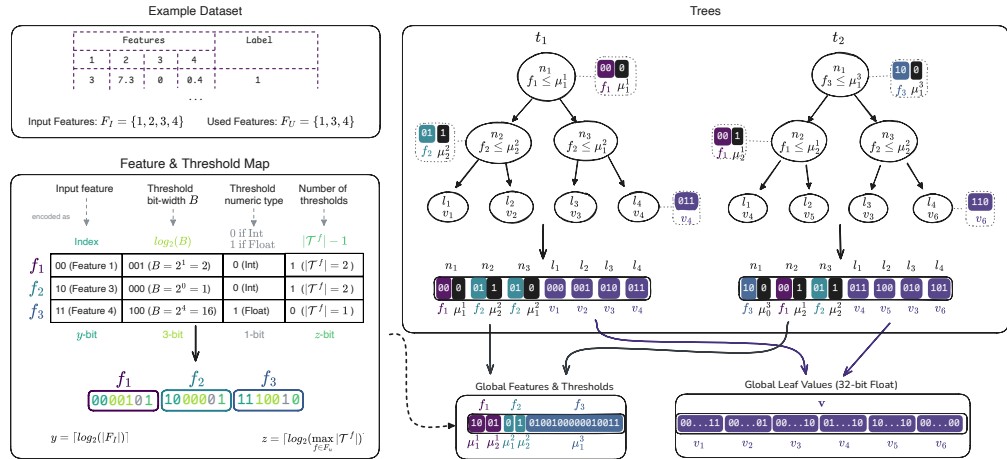

Figure 3: Illustration of a bit-wise encoding of a boosted tree ensemble with two trees. Each tree is stored in a bit-wise manner, with each internal node storing a reference to a feature index and a reference to a threshold index. For instance, for the left child $n_2$ of the root of the tree $t_1$, a reference to feature $f_2$ is stored along with a reference to the associated threshold $\mu_2^2$. For feature $f_2$, there are two thresholds used by the entire ensemble, namely $\mu_1^2$ and $\mu_2^2$, which can be used by any node of any of the trees (e.g. $t_2$ in node $n_3$). Accordingly, the leaf values stored in the leaves of the trees are shared (e.g. $v_4$ is used in leaf $l_4$ of tree $t_1$ and leaf $l_1$ of $t_2$) and stored in one array. Since the bit-size of the thresholds varies, additional metadata is stored in the `Feature & Threshold Mapping` table/array. For instance, there are two thresholds for feature $f_1$ of bit-size 2 (i.e., four different values), whereas there are two 1-bit thresholds for feature $f_2$.

- *Internal nodes:* For each internal node $n_i$, two pieces of information are stored, namely a reference for the feature $f_i \in F_U$ that is used for splitting along with an index for the associated threshold value $\mu_j^i$ ($j$-th threshold associated with feature $f_i$). For instance, for tree $t_2$ and node $n_1$, we store a reference `10` for feature $f_3$ and an index `0` for the associated threshold $\mu_1^3$. The reference `10` can be used to loop up the relevant information in the `Feature & Threshold Map`. In this case, $f_3$ corresponds to the fourth feature, $2^4$ bits are used to represent the thresholds (floating point), and one threshold was used overall for that feature. Using the index `0` and along with the information stored for the other features, one can then loop up the threshold value $\mu_1^3$.

- *Leaf nodes:* For each leaf node, a reference to its leaf value $v$ is stored. For instance, for the leaf $l_1$ of tree $t_1$, a reference `000` to the leaf value $v_1$ is stored. All these leaf values are shared across all the trees and are stored in the array `Global Leaf Values`.

The details for each feature are stored in a bit-wise manner in the array `Feature & Thresholds Map` that is referred to when decoding the nodes. To process such a reference we need to know the bit widths of following information:

(a) `Input feature index`: For a given dataset with features $F_I = \{1, \ldots, d\}$, the number $d$ of features is known and can be encoded using $\lceil \log_2(|F_I|) \rceil$ bits.

(b) `Threshold bit-width`: Threshold values are assumed to be representable as 1-bit (binary feature), 2|4-bits (small integers), or 8|16|32-bits (floating point with different precision or integers). The threshold bit-width per feature can be stored as a power of two, requiring only three bits to represent the aforementioned values ($2^0$ to $2^5$).

(c) `Threshold numeric type`: The representation can be either a floating point number (float) or a fixed point number (integer), and can be stored using a single bit, allowing for big integers and floating point values to be used.

(d) `Number of thresholds`: The maximum number $\max_{f \in F_U} |\mathcal{T}^f|$ of thresholds among all features can be determined at training time and can be encoded using $\lceil \log_2(\max_{f \in F_U} |\mathcal{T}^f|) \rceil$ bits. Since features with 0 thresholds are not included, we map the value 0 to the threshold count 1 (i.e., the bit value $+1$ is the actual count).

Hence, the array-based representation `Trees` of the trees $t_1, \ldots, t_K$ along with the `Feature & Threshold Map` are used to store the trees and additional information to retrieve the actual split feature indices and thresholds and leaf values in the two global arrays, which are described next.

### 3.2.2 SHARING THRESHOLDS AND LEAF VALUES

The threshold values are stored on a per-feature basis in a single array, see `Global Features & Thresholds` in Figure 3, which is referenced by nodes throughout the entire tree ensemble. For example, consider a tree node $n_1$ referencing the first feature, $f_1$, along with its corresponding first threshold, $\mu_1^1$, as depicted in tree $t_1$ in Figure 3. The `Feature & Threshold Map` allows for calculating the offset for each feature by determining the memory consumption of all previous features. Therefore, the associated threshold value (i.e. `10` for $\mu_1^1$) can be extracted and decoded to its original representation (i.e. $10 \rightarrow (int)2$ for $\mu_1^1$). This permits the variation of both the bit size and precision between different features within a single array.

The leaf values are stored (globally) in the array `Global Leaf Values` using a fixed 32-bit floating point representation. This allows for a high precision in leaves and a reuse across the different trees in the ensemble without feature reference. For example, consider the fourth global leaf value $v_4$ in Figure 3, which is referenced by both the leaf $l_4$ in tree $t_1$ and leaf $l_1$ of $t_2$.

## 4 EXPERIMENTS

To assess the quality of our compression approach and the impact of the additional parameters on the results, we ran three kinds of experiments: (1) A performance comparison of `ToaD` with other GBDT and tree ensemble optimization methods, (2) an univariate sensitivity analysis evaluating the threshold and feature penalties independently, and (3) a multivariate analysis combining both penalties. For the evaluation, the `ToaD` models were trained with varying hyperparameters (grid-search). The maximum number of iterations ranges from $2^0$ to $2^{10}$, maximum depth per tree from $2^0$ to $2^3$, and $\iota$ and $\xi$ from $2^{-10}$ to $2^{15}$. Moreover, $\iota$ and $\xi$ were set to 0 in every possible combination. This results in 32,076 models trained per dataset.

### 4.1 IMPLEMENTATION DETAILS

Our implementation builds upon the LightGBM framework[4]. The penalties were added as optional hyperparaneters to the training process of GBDT. The parameter $\iota$ is introduced as variable `toad_penalty_feature` and $\xi$ as variable `toad_penalty_threshold`. Moreover, logging the used features and thresholds enables tracking of the memory consumed by the selected memory layout. Therefore, the optional variable `toad_forestsize` allows training models for a specific memory limitation (such as 32KB on an Arduino Uno Rev 3). Experiments were conducted on a collection of eight widely used publicly available datasets (see Appendix B for specifics). We split all datasets into training and test sets using an 80/20 ratio. For the two smallest datasets, i.e. Breast Cancer and kr-vs-kp, we used 5-fold cross-validation on the training data, while for bigger datasets we used 10% from the training data as validation data. Model fitting was conducted on the training set, and the test was used to measure the final induced quality of the models. As metrics for quality measurement of the resulting models, accuracy is used for classification datasets and the $R^2$ score for regression datasets (Lewis-Beck & Lewis-Beck, 2015). Note that, for both metrics, higher values indicate better model performance.

### 4.2 MODEL COMPARISON TO BASELINES

We compared the performance of our approach to that of other efficient GBDT (compression) methods. LightGBM is considered an established framework for training boosted trees (Ke et al., 2017). It was used for comparison in both the standard and quantized version. For quantization, the threshold and leaf values were reduced to 16-bit floating point precision. In addition, LightGBM was evaluated in an array-based structure, i.e., it was stored without pointers but assuming all trees are complete as described in Section 3.2.1. This allows the comparison between `ToaD` and

---

[4]All the source code and the experimental setup are available at the repository: `https://github.com/TinyAIoT/LightGBM-ToaD`

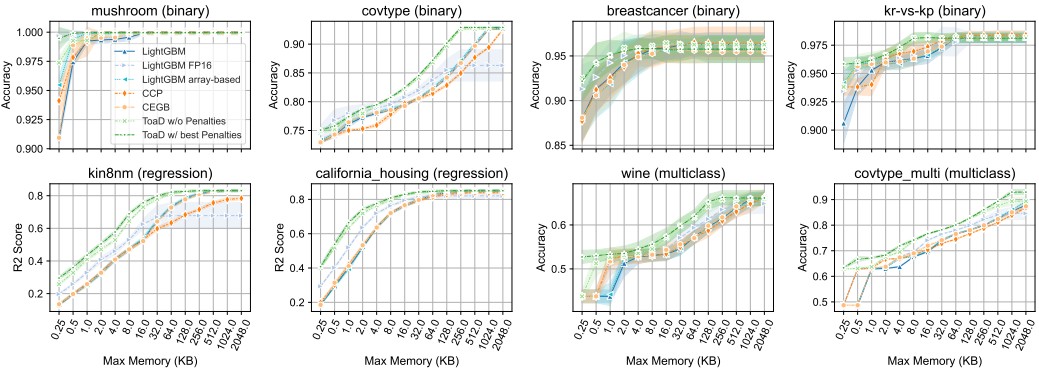

Figure 4: Accuracy vs. memory (KB) for `ToaD` and baselines. All models were trained on 12 train-test splits and the best model performance at a given memory limit from the hyperparameter analysis is depicted. The points show the mean across the splits, the errorbars show the respective standard deviation.

LightGBM under a unified pointer-less layout. Moreover, different pruning methods were evaluated with cost-efficient gradient boosting (CEGB) (Peter et al., 2017) and minimal cost-complexity pruning (CCP) (Breiman et al., 1984). All models were trained with the same hyperparameters as the `ToaD` models, i.e. all combinations of $2^0$ to $2^{10}$ maximum trees and maximum tree depth between $2^0$ and $2^3$. For all datasets we used $1-12$ as random seeds for the data split. Alongside related work (e.g., Buschjäger & Morik (2023)), we calculated the memory usage of a model with 128 bits per node, assuming all values are stored in single precision (float32) and 64 bits for quantized half-precision models.[5] In contrast to Buschjäger & Morik (2023), we assume the information about a node being a leaf can be encoded by a specific feature and child node identifier, thus no additional boolean values are required. Moreover, boosted trees do not need to store the class information within a leaf but create one ensemble per class for multiclass classification problems.

For the calculation of the memory footprint of the `ToaD` models, the proposed memory layout is used. For the `ToaD` models, it is distinguished between the memory layout without applying penalization during training, i.e., $\iota = 0$ and $\xi = 0$, and the best-performing models with penalization. For the comparison visualized in Figure 4, the best-performing models with a memory consumption less than or equal to the respective upper limit were chosen from the grid search results. Thus, the number or depth of trees of models at the same memory limit for the same dataset may differ for different model types. We expect `ToaD` to outperform the baseline implementations as it does not require pointers to its children, it encodes boolean values with only 2 bits, and a suitable penalty configuration encourages the model to reuse features and thresholds. Thus, `ToaD` especially benefits from the pointer-less tree layout, which is strong for complete trees, whereas a standard child-pointer layout also allows deeper non-complete trees without wasting too many resources.

### 4.2.1 RESULTS BASELINE COMPARISON

Since the primary use case for `ToaD` is microcontrollers, typical memory limits are considered for the respective performance comparison. The results are depicted in Figure 4. For almost all investigated configurations, `ToaD` outperforms the baseline approaches, even more when considering the models from the penalized training. On all tested multiclass datasets, the `ToaD` approaches are superior to the other models across all memory limits. The same holds for the regression datasets until the performance saturates, starting with different memory limits, with some of the competitors catching up to the same score with increasing memory available. In the

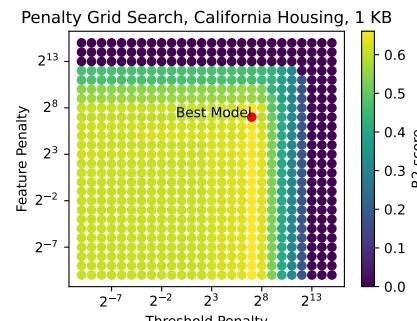

Figure 5: Model performance on California Housing with a 1 KB memory limit under varying penalties.

---

[5]Each node stores four values: one feature identifier, one threshold, and two child pointers.

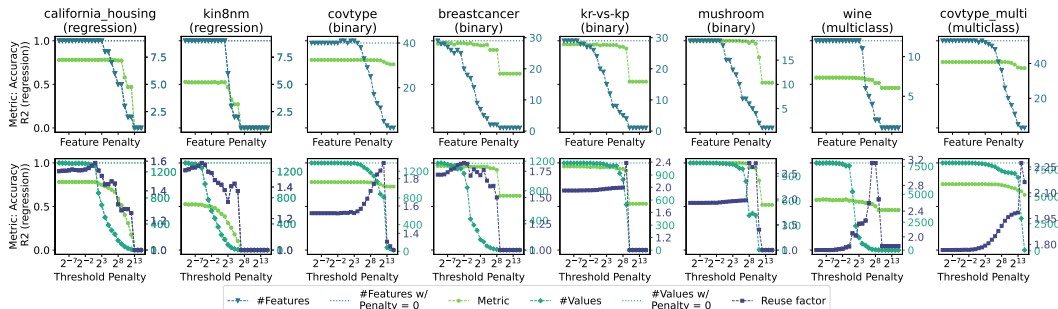

Figure 6: Influence of penalties $\iota$ and $\xi$ on the number of thresholds and the number of features, displayed on the right y-axis, alongside the respective performance scores shown on the left y-axis. The threshold penalty figure additionally depicts the reuse factor. The maximum number of iterations per model is set to 256 during training with a maximum tree depth of 2.

interesting memory range up to 128 KB, before some model performances saturate, the competing methods need 4 to 16 times the memory to achieve the same performance. For example, on the Covertype multiclass dataset, the best `ToaD` model at 2 KB achieves an accuracy of 69 %, which quantized LightGBM as the best competitor only matches with 8 KB, while float32 LightGBM even needs 16 KB. A further finding includes that `ToaD` outperforms array-based LightGBM, showcasing the effectiveness of our approach beyond the pure pointer-less layout.

Figure 5 shows an exemplary grid for model performances at a given memory limit for the California Housing dataset. The maximum memory size is fixed, allowing for an unlimited number of trees and nodes. As the `forestsize` parameter determining the memory limitation can be set by the user within our implementation, this graph can easily be generated for any memory size to determine the best penalty configuration. This approach helps identify the best-performing model for a given dataset on memory-limited hardware.[6]

## 4.3 UNIVARIATE SENSITIVITY ANALYSIS

We conducted a univariate sensitivity analysis of the two newly introduced penalties to assess their individual effects on the model. During training, we varied the feature penalty $\iota$ and the threshold penalty $\xi$ independently over the range $2^{-10}$ to $2^{15}$, setting the other parameter to zero. We then tracked the number of nodes and leaves, the number of global values (thresholds and leaf values), and the test set performance. Furthermore, to evaluate how efficiently threshold values were reused, we calculated the reusing factor ($ReF$) as the ratio between the sum of the nodes and leaves and the global number of values. In a naive implementation, the number of leaves and nodes equals the count of values, resulting in $ReF = 1$. With the ratio $ReF$, we can determine how many of these values are reused by the `ToaD` approach. For instance, a value of $ReF = 1.5$ can be interpreted as a model reusing 50% of its threshold and leaf values, while $ReF = 2.0$ indicates that, on average, each value is used twice. Results of this analysis are depicted in Figure 6 with a maximum number of iterations of 256 and a maximum depth per tree of 2, as `ToaD` is meant to be especially useful for shallow trees. A selection of results for further hyperparameter settings can be found in Appendix E.2.

### 4.3.1 RESULTS FEATURE PENALTY

Figure 6 (top) shows the univariate sensitivity analysis for the feature penalty $\iota$. For $\iota < 1$, the number of features is largely unchanged, except for a notable drop in the Breast Cancer dataset. The value of the feature penalty at which it takes effect varies between different datasets. For datasets with few features (California Housing and kin8nm), the accuracy drops shortly after the number of features decreases. We assume the few features in this dataset are essential for accurate predictions. In contrast, datasets with more features show a slower and later accuracy decline, as the penalty first removes less relevant features. For example, the Covertype model loses only $\approx 2\%$ accuracy when $\iota = 2^{12}$, while the feature count drops from 35 to 5.

---

[6]A script to reproduce the figures is available in the repository.

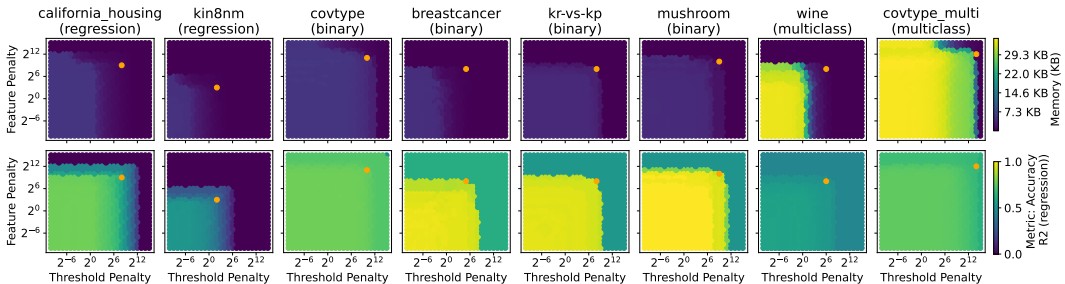

Figure 7: Influence of $\iota$ and $\xi$ on the needed memory (KB) and the representative metric (accuracy or $R^2$ score). Orange dots mark penalty configuration combinations that are a good trade-off between model accuracy and memory. The maximum number of iterations per model is set to 256 during training with a maximum tree depth of 2.

### 4.3.2 RESULTS THRESHOLD PENALTY

In the bottom row of Figure 6, the performance metric for the model, the number of global values, and the $ReF$ are depicted against a varying threshold penalty $\xi$. For all datasets, an increase in the threshold penalty decreases the number of global values used by the model. For the maximum penalty of $\xi = 2^{15}$, the number of global values approaches 1 for all models, i.e., the model only consists of one tree with the root node. The trend of the performance metrics differs between the datasets. The accuracy drops abruptly after a certain penalty for most of the binary datasets, whereas the performance of the other datasets declines more gradually. The models trained on the Covertype dataset record the slightest decrease of about 6% from the smallest to the largest penalty value. At the same time, the number of values used for the model drops from 1323 to only 18. These differences probably arise from the size of the datasets, as the more stable models have more data points to choose from.

The reuse factor follows a similar pattern across all datasets. At first, it increases for higher penalties, but with very high penalties, it starts to decrease abruptly. At its peak, every model achieves a $ReF$ of at least 1.5 with the Wine dataset reusing each value more than three times for a penalty around $2^8$. In contrast, for the highest investigated threshold penalty of $\xi = 2^{15}$, all models – except for the Covertype dataset – result in $ReF = 1$, meaning each value is only used once per feature. This decrease in $ReF$ for very high $\xi$ values can be explained by fewer global values being available for reuse, thus reducing reuse opportunities across training splits. Interestingly, the $ReF$ drops later than the number of values; actually, first it increases, and with the sharp decrease of the $ReF$, also the accuracy plunges. Of special interest for the training process are the penalties with $ReF$ peaks, as here the model performance is still satisfactory, but the number of global thresholds is low.

### 4.4 MULTIVARIATE SENSITIVITY ANALYSIS

To assess the combined effects of our penalties, we performed a multivariate sensitivity analysis for both penalty parameter values. As in the univariate sensitivity analysis, we used the parameter space $2^{-10}$ to $2^{15}$ for both parameters, resulting in $26 \times 26 = 676$ trained models per dataset and tree setting. Figure 7 shows the memory consumption next to the performance metric in a grid of the combined penalties with a maximum of 256 iterations with a maximum tree depth of 2. A selection of results for further hyperparamter settings can be found in Appendix E.3.

The memory requirements for all datasets decreased significantly as penalties increased, with each dataset having a specific threshold at which memory usage drops rapidly. The different looks of the multiclass dataset memories are reasoned by them using one tree ensemble per class; thus, more trees and memory are needed. For the larger datasets (Covertype, California Housing), the difference in memory consumption is stronger, starting at around 5 KB for small penalties and dropping to around 80 Byte for larger penalties. On average, predictive performance is better for smaller penalties as more features and thresholds are used. Independent of the dataset and objective, after a certain threshold in the feature penalty, model predictions are not better than guessing. This is expected as the penalties reduced tree complexity, thus it loses its predictive capabilities with the omitted features and thresholds.

By combining both penalty parameters, we can select a model that maintains high predictive performance while having a significantly smaller memory footprint. This creates solutions that are equally viable, known as nondominated solutions (Deb, 2011), where no solution is better in both predictive performance and memory usage simultaneously. If we know the minimal predictive performance we need for a task, we can select the corresponding microcontroller accordingly. In Figure 7, exemplary points are marked in orange where the accuracy is good, but the memory usage dropped.Notably, the ideal combination of the hyperparameters highly depends on the use-case. As the objectives correlate negatively, solely $3.37\%$ of the solutions are dominated, leaving many possible options. One is advised to limit the search space to a reasonable number of nodes and trees and, firstly, increase both penalties similarly, as good solutions seem to appear more often in this space. Findings from our intensive hyperparameter grid search suggest that the penalties can be set to values greater than $2^0 = 1$. For bigger datasets or multiclass classification values starting from $2^5 = 32$ seem to work well. Thus the search space and with this also the computational demand can be reduced significantly compared to the conducted experiments. However, this advise has to be taken with caution as datasets with few features or few thresholds might show other patterns.

To summarize, there is no globally optimal penalty setting as it depends on the dataset being used and the number and depth of trees allowed during training. For all datasets, we found that higher memory usage does not necessitate an incline in predictive performance. Instead, the metrics remained similar until there was a considerable decrease in memory usage.

## 5 CONCLUSION & FUTURE WORK

We propose two hyperparameters and a new memory layout to optimize the memory footprint of boosted decision tree ensembles. First, custom penalties within boosted decision tree training were implemented that encourage a boosted tree to reuse features and thresholds and thus create smaller models. An univariate analysis has shown an effective decrease in the number of utilized values but almost unchanged performance for specific penalty values. Then, the decision tree memory layout `ToaD` was introduced. Building upon index-based trees that enable a pointer-less node sequence and global value lookup, it allows the reuse of threshold values multiple times per feature and allows storing thresholds with fewer bits, e.g. only 1 bit for boolean values. Our experiments show that we can store models with a significantly smaller memory footprint than baseline methods while maintaining the same accuracy, supporting the application of powerful boosted decision trees on resource-constrained devices.

Although the linear penalizer already performed well, a deeper analysis of more sophisticated penalizers may reveal even better performance and thus would prove to be an interesting extension to the present work. Adapting our method to reuse leaf values more effectively could also prove useful as well as the transfer to other variants of decision tree ensembles. Lastly, to further assess the effectiveness of the proposed method, its deployment to different microcontroller units would be valuable.

### REPRODUCIBILITY STATEMENT

To ensure a high usability and reproducibility of our findings we implemented the approach on top of the frequently used LightGBM framework. The respective implementations and code to rerun experiments can be found at: `https://github.com/TinyAIoT/LightGBM-ToaD`. To ensure reproducibility we refer to the release `v1.0.0`. Our evaluation partly relies on the Python package `lightgbm` version 4.6.0. The references for the datasets used for the experiments can be found in Appendix B, while the random seeds used for splitting are documented in Section 4.2.

### ACKNOWLEDGMENTS

This work was funded by the German Federal Ministry for the Environment, Nature Conservation, Nuclear Safety and Consumer protection, Project TinyAIoT, Funding Nr. 67KI32002A. Calculations for this publication were performed on the HPC cluster PALMA II of the University of Münster, subsidised by the DFG (INST 211/667-1).

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

# A    TRAINING COMPRESSED ENSEMBLES

Constructing an optimal tree ensemble $T$ w.r.t. Equation (1) is computationally infeasible. There-fore, one resorts to $K$ boosting rounds. In each boosting round $m$, one considers $T_{m-1} := \sum_{k=1}^{m-1} t_k$ together with a new tree $t_m$ that is built in this round (starting with $T_0 := 0$) (Chen & Guestrin, 2016). The new tree is chosen such that

$$\sum_{i=1}^{n} \mathcal{L}\left(y_i, T_{m-1}(\mathbf{x}_i) + t_m(\mathbf{x}_i)\right) + \sum_{k=1}^{m} \Omega(t_k) \tag{4}$$

is minimized. A common choice for the regularizer $\Omega$ is

$$\Omega(t_m) = \gamma L + \frac{1}{2}\lambda \sum_{j=1}^{L} v_j^2,$$

where $L$ is the number of leaves and $v_1, \ldots, v_L \in \mathbb{R}$ are the associated leaf values of $t_m$. This penalizes trees with many leaves and large absolute leaf values. The modified regularizer

$$\Omega_l(t_m) = \Omega(t_m) + \iota \cdot |F_U| + \xi \cdot \sum_{f \in F_U} |\mathcal{T}^f|, \tag{5}$$

proposed in Section 3, with user-defined hyperparameters $\iota, \xi \in \mathbb{R}^+$, penalizes the use of new features and thresholds in a linear manner. Similar to Peter et al. (2017), incorporating $\Omega_l$ into the objective leads to a modified gain compared to standard boosted decision trees (Chen & Guestrin, 2016). More precisely, using gradient statistics of the form

$$g_i := \left. \frac{\partial}{\partial z} \mathcal{L}(y_i, z) \right|_{z=T_{m-1}(\mathbf{x}_i)}$$

$$h_i := \left. \frac{\partial^2}{\partial^2 z} \mathcal{L}(y_i, z) \right|_{z=T_{m-1}(\mathbf{x}_i)}$$

and omitting constant terms, one obtains the simplified objective

$$\sum_{j=1}^{L} \left[ G_{I_j} v_j + \frac{1}{2} \left( H_{I_j} + \lambda \right) v_j^2 \right] + \gamma L + \iota |F_U| + \xi \cdot \sum_{f \in F_U} |\mathcal{T}^f| \tag{6}$$

with $G_S := \sum_{i \in S} g_i$ and $H_S := \sum_{i \in S} h_i$. Here, $I_j$ denotes the set of training indices assigned to the $j$-th leaf of the current tree $t_m$, i.e., $I_j = \{i \mid q(\mathbf{x}_i) = j\}$, where $q : \mathbb{R}^d \to \{1, \ldots, L\}$ maps an input instance to the corresponding leaf of $t_m$ (i.e., $t_m(\mathbf{x}) = v_{q(\mathbf{x})}$).

The decision tree $t_m$ is constructed greedily by iteratively splitting leaves (starting at the root) if this improves Objective (6); otherwise, the construction stops. More specifically, given a leaf associated with a set $I$ of training indices, a split on feature $i \in \{1, \ldots, d\}$ with threshold $\mu$ divides $I$ into two subsets, $I_L$ (left leaf) and $I_R$ (right leaf). The corresponding gain is then given by

$$\Delta_l(I, i, \mu) := \frac{1}{2} \left( \frac{G_{I_L}^2}{H_{I_L} + \lambda} + \frac{G_{I_R}^2}{H_{I_R} + \lambda} - \frac{(G_I)^2}{H_I + \lambda} \right) - \gamma - s_f \iota - s_t \xi, \tag{7}$$

$$= \Delta(I, i, \mu) - s_f \iota - s_t \xi, \tag{8}$$

where $s_f = 1$ if a new feature is used (and $s_f = 0$ otherwise), and $s_t = 1$ if a new threshold is used (and $s_t = 0$ otherwise). Thus, the modified regularizer (5) introduces the additional terms $-s_f \iota$ and $-s_t \xi$, corresponding to the cost of using a new feature or threshold, respectively.

## B    DATASETS

An overview of the eight datasets used for experiments in Section 4 along with their statistics are presented in Table 1. The binary Covertype (`covtype`) dataset (Blackard, 1998), the California Housing (`california_housing`) regression dataset (Kelley Pace & Barry, 1997), the KRKPA7 (`kr-vs-kp`) dataset (Shapiro, 1983) and the Breast Cancer (`breastcancer`) dataset (Wolberg et al., 1993) are commonly solved using boosted decision trees. The `kin8nm` dataset (Ghahramani), which describes robotics decision making, and the Mushroom (`mushroom`) dataset (Mushroom, 1981), which is for mushroom edibility classification, cover data that might be of interest for analysis on constrained edge devices.

Table 1: Datasets

| Dataset | Instances | Features | Task |
|---|---|---|---|
| Covertype (Blackard, 1998) | 581,012 | 54 | Binary & multiclass classification |
| California Housing (Kelley Pace & Barry, 1997) | 20,640 | 8 | Regression |
| kin8nm (Ghahramani) [7] | 8,192 | 8 | Regression |
| Mushroom (Mushroom, 1981) | 8,124 | 22 | Binary classification |
| Wine Quality (Cortez et al., 2009) | 6,497 | 11 | Multiclass classification |
| KRKPA7 / kr-vs-kp (Shapiro, 1983) | 3,196 | 36 | Binary classification |
| Breast Cancer Wisconsin (diagnostic) (Wolberg et al., 1993) | 569 | 30 | Binary classification |

## C    EXTENDED RELATED WORK

Decision trees and decision tree ensembles, such as GBDT, have remained very popular, despite the rise of deep learning models (Grinsztajn et al., 2022). These tree-based methods are highly valued for their interpretability, particularly when dealing with small datasets, mixed data types, and constrained computational resources. Recent advances in tree-based models have also introduced several approaches to enhance their efficiency. For instance, cost-efficient gradient boosting introduces mechanisms to account for feature acquisition and tree evaluation costs (Peter et al., 2017), which involves penalizing the use of new features based on their cost and minimizing the number of split nodes an input traverses during inference. Other work (Ponomareva et al., 2017) presents a method to train compact boosted tree ensembles for multi-class classification using vector-valued trees and layer-by-layer boosting. Bonsai (Kumar et al., 2017), instead, yields a single decision tree model and reduces the model size by projecting the input data into a low-dimensional subspace. Two other approaches are QuickScorer (Lucchese et al., 2017) and its enhanced version, RapidScorer (Ye et al., 2018), decision tree ensembles both developed for search engines, where array structures are used to reduce model size. These methods prioritize enhancing processing speed over minimizing memory footprint by summarizing features and split values for joint calculation. Recently, Koschel et al. (2023) further adapted these approaches to IoT contexts, focusing on ARM CPUs prevalent in such devices. Their work involves adapting QuickScorer variants for these CPUs and applying fixed-point quantization to split nodes and leaf values, emphasizing computational enhancements over memory optimization. DimBoost (Jiang et al., 2018) made use of lower precision values during the calculation of gradient histograms in training as one of their contributions to improve the performance of GBDT for high-dimensional data. Additionally, Shi et al. (2022) demonstrated that using just two or three bits suffices to represent gradients in GBDT training. Devos et al. (2020) represent input data as well as gradients in bit-level data structures to improve the runtime of GBDT training.

The related works sketched above aim mostly at enhancing the training process, leading to improvements in memory and energy consumption as ancillary benefits. The notable exception are the works by Kumar et al. (2017) and Ponomareva et al. (2017), which explicitly address a memory footprint reduction in the context of resource-constrained devices, achieving compact and effective tree-based

---

[7] https://www.cs.toronto.edu/~delve/data/kin/desc.html

models. However, the approach of Kumar et al. (2017) only considers single decision trees and conceptually modifies the tree building process and the underlying models. Ponomareva et al. (2017) focus on multi-class classification problems. However, the vector-values used within trees may increase the memory footprint of tree models for binary classification or for low-number multi-class problems. Additionally, an evaluation of memory consumption is not conducted directly, but rather the number of trees is used as an estimate. Specific memory consumption of different Machine Learning (ML) models and compression techniques is considered by Buschjäger & Morik (2023) and (Devos et al., 2025). Their works combine ensemble pruning and update of leaves values as post-training compression methods to reduce the size of RFs and GBDTs respectively. Lossless post-training pruning of boosted ensembles is proposed by Emine et al. (2025). Liu & Mazumder (2023), on the other hand, focus on pruning deep layers of trees to tackle growing memory demands because of the exponential increase in the number of nodes for deeper layers. There are many other works optimizing non-decision tree-based methods for deployment on resource-constrained devices, for example, based on k-nearest neighbors Gupta et al. (2017). In this work, we focus on gradient boosted trees, and at minimizing the memory footprint of the overall ensemble in the course of the training process. We extend the work of Peter et al. (2017) by defining feature costs that aim at reducing the bits required to encode feature indices. We also introduce a special cost for adding new thresholds to each feature and provide a specialized encoding for the induced binary trees, including global threshold arrays that are shared across all individual learners.

## D    COMPARISON TO RANDOM FOREST

`ToaD` as a compression method developed for GBDT is compared to a baseline RF method for classification tasks, as well as version pruned along Guo et al. (2018), as shown in Figure 8. This comparison is limited to a maximum of 256 trees; otherwise, the same hyperparameters as before have been used (Section 4.2). Only classification results are included, since the used pruning method is not designed for regression tasks. RF methods can have a slight performance advantage for multiclass problems, since boosted trees are usually trained with one ensemble per class, whereas a RF stores the class information in the nodes. An extension of our work in the future could include optimizing `ToaD` for multiclass classification, for example, based on the approach of Ponomareva et al. (2017).

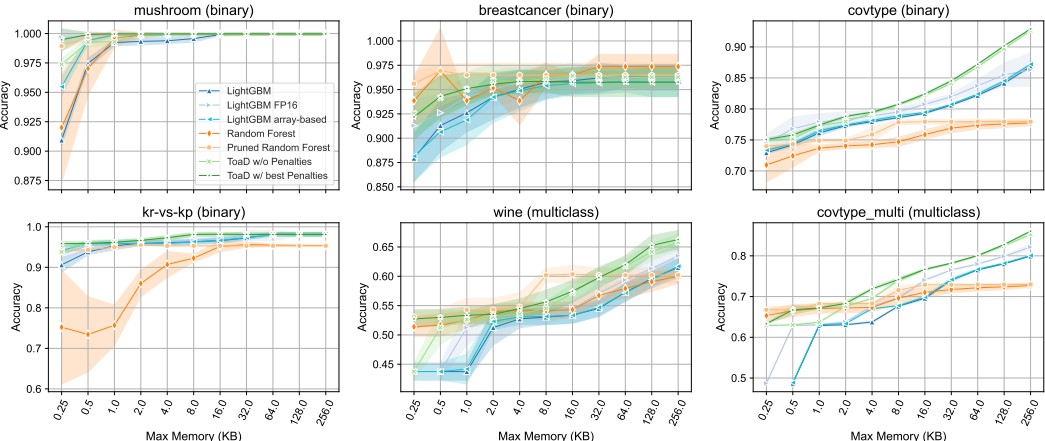

Figure 8: Comparison of baseline LightGBM methods and `ToaD` to baseline RFs and RFs pruned by the method presented by Guo et al. (2018). The models were trained using 12 train-test data splits.

# E EXPERIMENTS

## E.1 RUNTIME EXPERIMENTS

For an estimate for the expected runtime, we deployed a `ToaD`-model processing the Covertype binary dataset at a memory limit of 0.5 KB. The model consists of four complete trees with a depth of four. We measured 20 runs, with each run having 500 predictions. We decided to stop at 20 runs as the variance between the runs was minimal. Experiments were conducted on the Seeed Xiao ESP32-S3 and the Arduino Nano 33 BLE Rev2. To ensure that programs are not further optimized, we used random numbers as input. The prototypes are available at the repository[8]. Note that currently the `ToaD`-program is only a first prototype, and there are many options for optimization. A single prediction took $0.51$ milliseconds for the `ToaD`-model on the Arduino Nano. In contrast to the LightGBM program, this is a slowdown by a factor of $\sim 5$. For the ESP32-S3, a single prediction took $0.14$ milliseconds, with a respective slowdown in contrast to LightGBM of a factor of $\sim 8$. Those findings should be read as an outlook for future work to incorporate optimization techniques to close the runtime gap. As already stressed in the main paper, the observed latency degradation is not a significant factor in real-world deployments. At below millisecond inference time for `ToaD`, the overall latency and energy consumption of real-world use cases is dominated by the device measuring the input data and potentially transmitting the results.

| Hardware | Average Prediction Runtime (μs) | |
|---|---|---|
| | ToaD | LightGBM |
| XIAO ESP32S3 | 137.08 | 17.63 |
| Arduino Nano 33 BLE | 512.89 | 102.16 |

Table 2: Runtime for one inference run for the Covertype binary dataset with four trees, each with a depth of four.

## E.2 UNIVARIATE SENSITIVITY ANALYSIS

Influence of feature penalty $\iota$ and threshold penalty $\xi$ on the number of thresholds and the number of features, for different hyperparamter settings are displayed in the following figures. The respective top row shows the influence of $\iota$ on the number of features and the performance when $\xi = 0$. The respective top row shows the influence of $\xi$ on the number of thresholds and the performance when $\iota = 0$. Additionally the reuse factor $ReF$ is displayed as described in Section 4.3.

We can observe similar patterns in the evolution of used features and thresholds and the corresponding performance as described in Section 4.3.

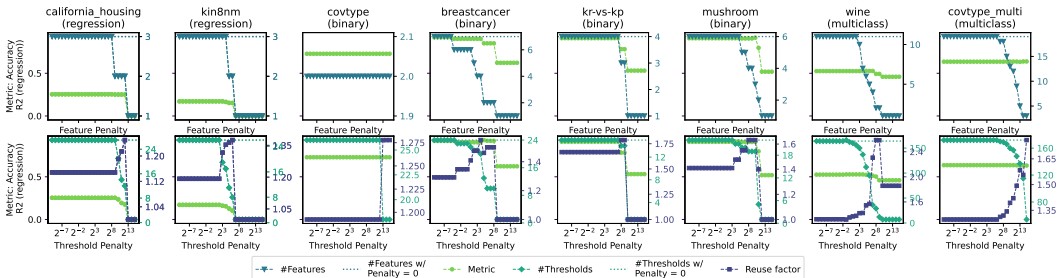

Figure 9: $max\_iterations = 4$, $max\_depth = 2$

---

[8] https://github.com/TinyAIoT/LightGBM-ToaD/tree/submission/experiments/latency

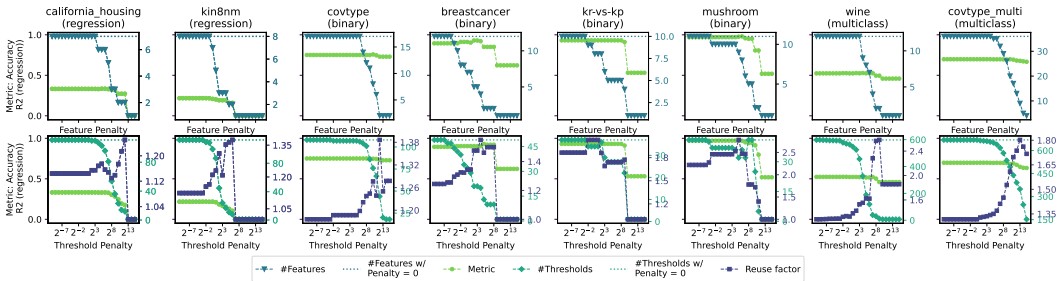

Figure 10: $max\_iterations = 4, max\_depth = 4$

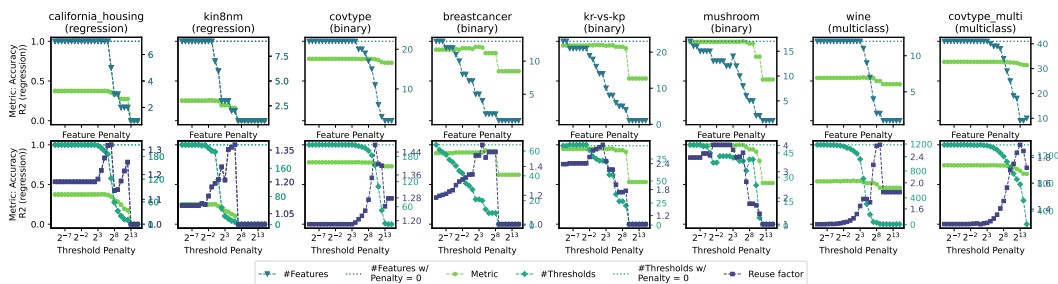

Figure 11: $max\_iterations = 4, max\_depth = 8$

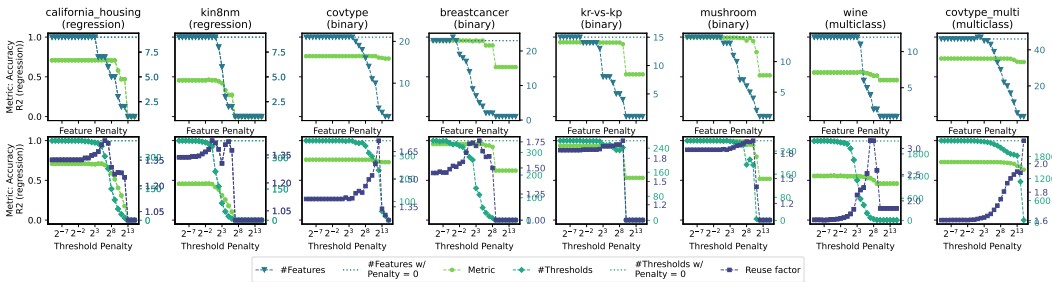

Figure 12: $max\_iterations = 64, max\_depth = 2$

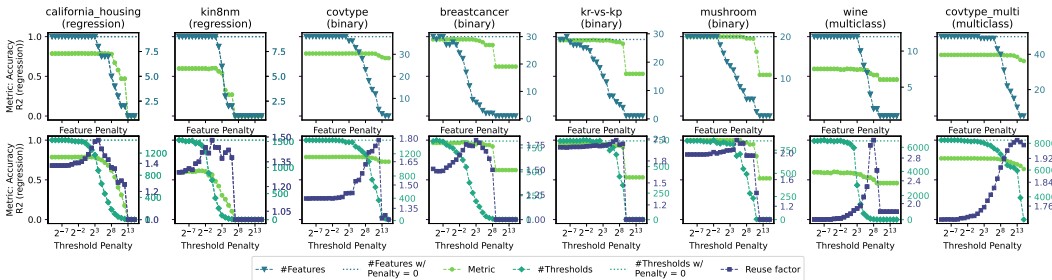

Figure 13: $max\_iterations = 64, max\_depth = 4$

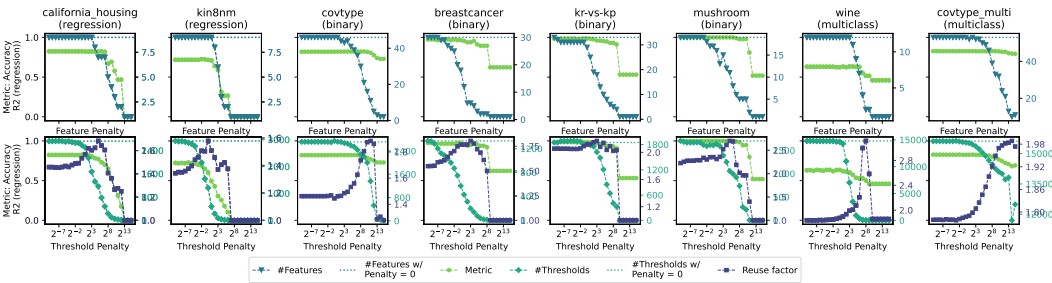

Figure 14: $max\_iterations = 64$, $max\_depth = 8$

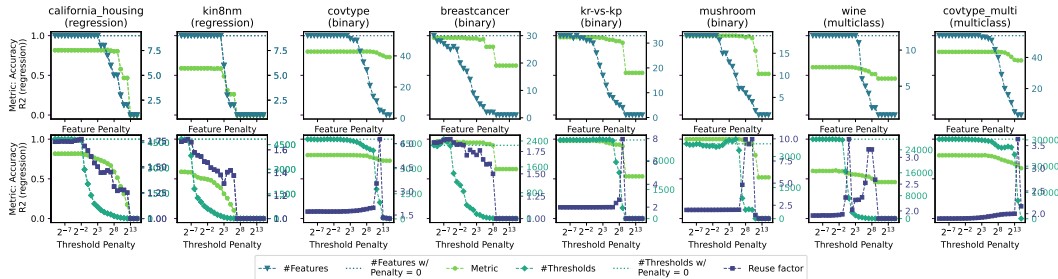

Figure 15: $max\_iterations = 1024$, $max\_depth = 2$

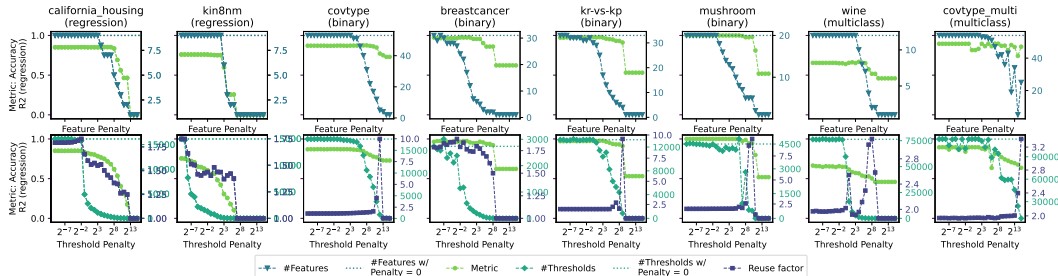

Figure 16: $max\_iterations = 1024$, $max\_depth = 4$

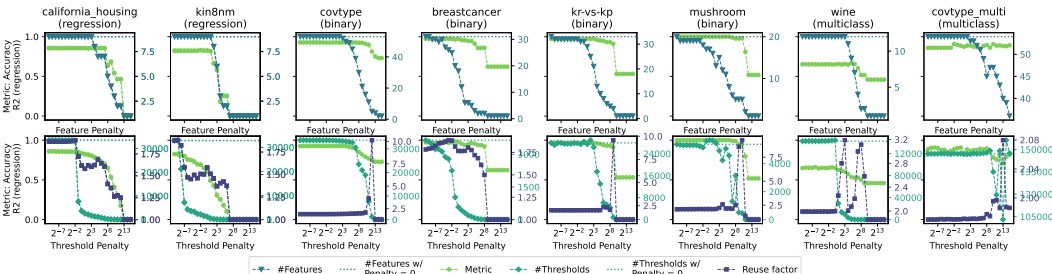

Figure 17: $max\_iterations = 1024$, $max\_depth = 8$

### E.3 MULTIVARIATE SENSITIVITY ANALYSIS

In the following figures the influence of $\iota$ and $\xi$ on the needed memory (KB) on the top row and the representative metric (accuracy or $R^2$ score) on the bottom row are depicted for different hyperparameter settings.

Similarly as depicted in Figure 7 and described in Section 4.4 we can find useful penalty combinations that provide a useful trade-off between a small decrease in performance but a significant decrease in memory consumption for most of the given examples.

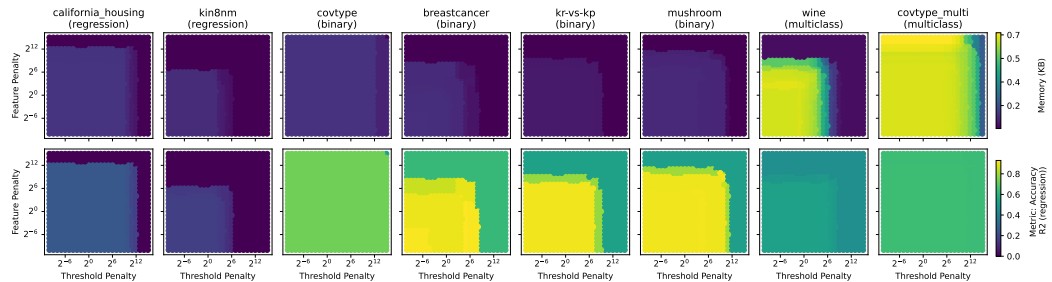

Figure 18: $max\_iterations = 4, max\_depth = 2$

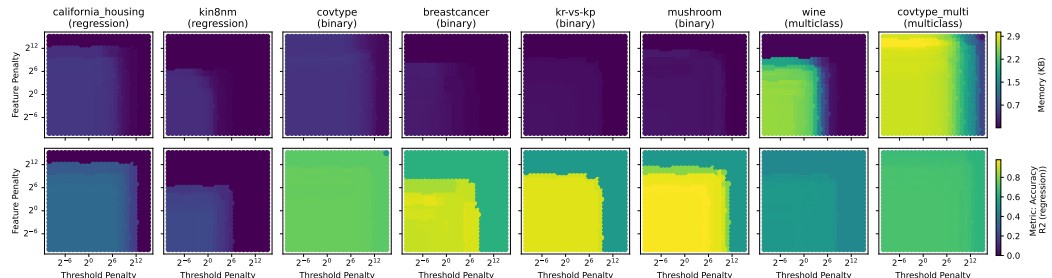

Figure 19: $max\_iterations = 4, max\_depth = 4$

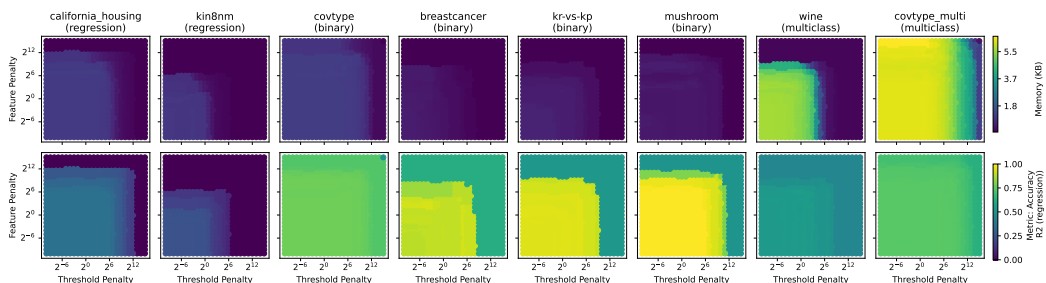

Figure 20: $max\_iterations = 4, max\_depth = 8$

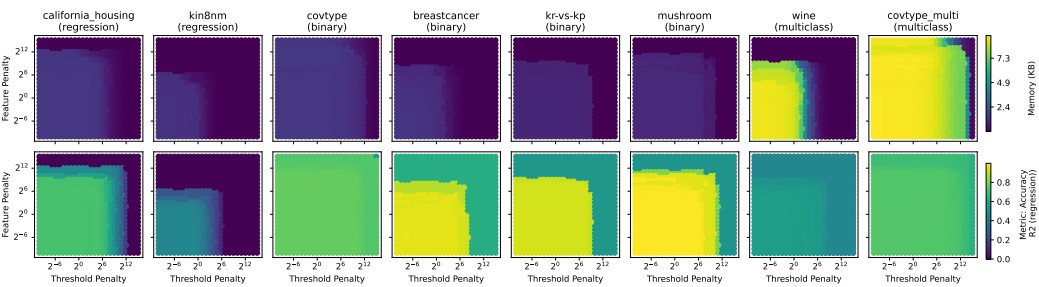

Figure 21: $max\_iterations = 64, max\_depth = 2$

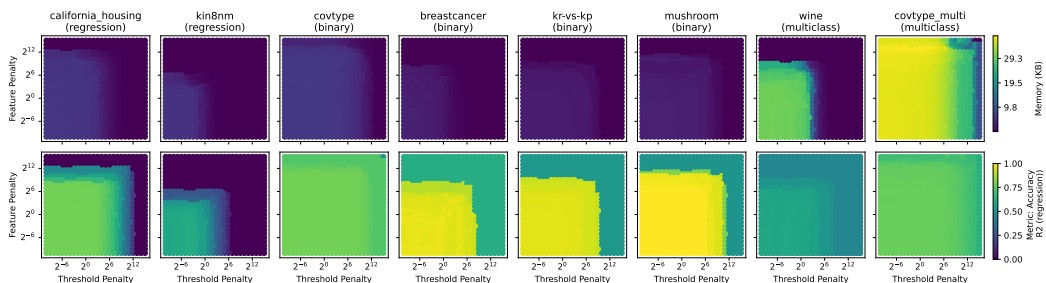

Figure 22: $max\_iterations = 64, max\_depth = 4$

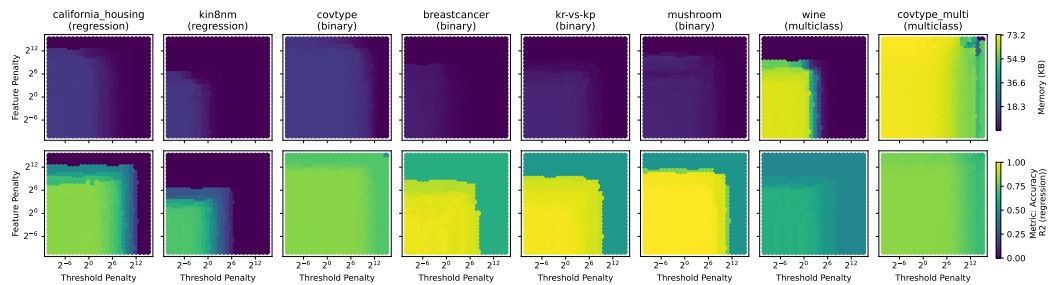

Figure 23: $max\_iterations = 64, max\_depth = 8$

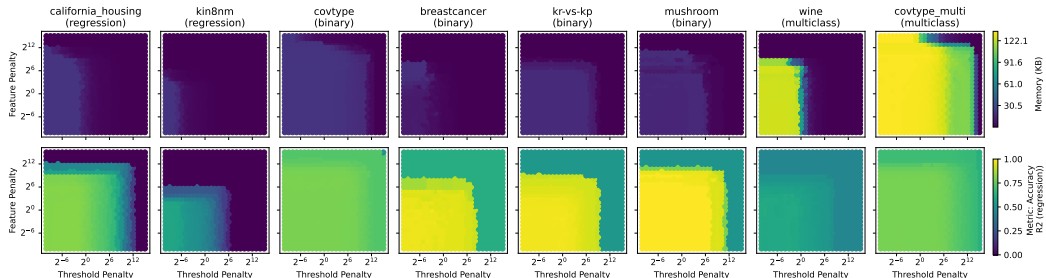

Figure 24: $max\_iterations = 1024, max\_depth = 2$

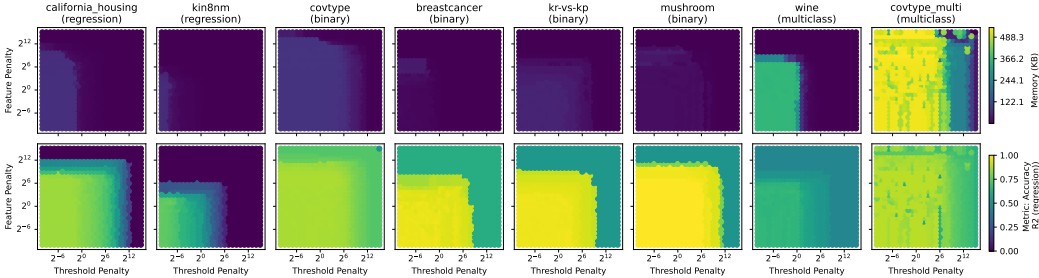

Figure 25: $max\_iterations = 1024, max\_depth = 4$

# F LARGE LANGUAGE MODEL USAGE

This manuscript has undergone sentence-level improvements using Large Language Models (LLMs) to enhance clarity and readability. However, all scientific ideas, methods, results and conclusions are the exclusive work of the authors.

