# OpenReview forum: "Boosted Trees on a Diet: Compact Models for Resource-Constrained Devices"
_ICLR.cc/2026/Conference — ICLR 2026 Poster_

### Official Review · Reviewer_gDno · 2025-10-30

**Soundness:** 3
**Presentation:** 3
**Contribution:** 2
**Rating:** 6
**Confidence:** 3

**Summary:**

This paper introduces compressed boosted decision trees for constrained devices. The authors reuse values across thresholds, leaves, etc. and create a compact memory layout. The results indicate that the proposed methods are competitive in accuracy while being significantly better in comression.

**Strengths:**

1. The paper is clear to read
2. The motivation and the constraints specified as per use cases and practical designs pique the interest
3. The proposed method is well-described and the method appears very intuitive
4. The results and analysis are rigorous and show good performance

**Weaknesses:**

1. I am not from this exact area but I am a bit surprised if there aer not more related works. Neural network pruning, quantization and deployment on edge devices is common. But, perhaps even for tree-based methods there are more such related work. Atleast, a couple of examples that come to my mind.

• ProtoNN: Compressed and Accurate kNN for Resource-scarce Devices
Chirag Gupta, Arun Sai Suggala, Ankit Goyal, Harsha Vardhan Simhadri, Bhargavi Paranjape, Ashish Kumar,
Saurabh Goyal, Raghavendra Udupa, Manik Varma and Prateek Jain
International Conference on Machine Learning (ICML), 2017

• Resource-efficient Machine Learning in 2 KB RAM for the Internet of Things
Ashish Kumar, Saurabh Goyal and Manik Varma
International Conference on Machine Learning (ICML), 2017


2. Continuation to above -- I think the absence of more baselines esp. in optimized tree implementations is a weakness.

**Questions:**

1. Can you expand related work to include more methods for optimizing tree-based methods on constrained devices?
2. Can you compare your work empirically to those papers?

---

> ### Author Response · Authors · 2025-11-21
> **Rebuttal - Part I**
>
> We appreciate your positive assessment of the problem’s relevance and the clarity of our method and presentation. Thank you as well for the constructive feedback on related work, which helped us strengthen the revision. We have tried to address your concerns below and are open for further feedback.
>
> ### Additional Related Works
>
> > W1: I am not from this exact area but I am a bit surprised if there are not more related works. Neural network pruning, quantization and deployment on edge devices is common. But, perhaps even for tree-based methods there are more such related work. Atleast, a couple of examples that come to my mind.
> > • ProtoNN: Compressed and Accurate kNN for Resource-scarce Devices Chirag Gupta, Arun Sai Suggala, Ankit Goyal, Harsha Vardhan Simhadri, Bhargavi Paranjape, Ashish Kumar, Saurabh Goyal, Raghavendra Udupa, Manik Varma and Prateek Jain International Conference on Machine Learning (ICML), 2017
> > • Resource-efficient Machine Learning in 2 KB RAM for the Internet of Things Ashish Kumar, Saurabh Goyal and Manik Varma International Conference on Machine Learning (ICML), 2017
>
> Thank you for pointing us to these related works. Bonsai (Kumar et al., 2017) was already included, as it also targets tree-based models for resource-constrained devices; however, it relies on a single tree, whereas our focus is on compressing boosted ensembles. ProtoNN, while conceptually more distant from our approach, is indeed relevant in the broader context of model compression for constrained hardware. We now reference it in the revised manuscript.
>
> > Q1: Can you expand related work to include more methods for optimizing tree-based methods on constrained devices?
>
> Thank you for the suggestion. We have expanded the related work section to include additional compression techniques for tree-based models. In particular, we have added four papers that focus on compressing random forests. Since random forests differ fundamentally from boosted trees (most notably in their typically deeper structures), we consider these approaches as not directly comparable to our setting. Nonetheless, we now include them for completeness and context.
>
> - B. Liu and R. Mazumder, “ForestPrune: Compact Depth-Pruned Tree Ensembles,” in Proceedings of The 26th International Conference on Artificial Intelligence and Statistics, F. Ruiz, J. Dy, and J.-W. van de Meent, Eds., in Proceedings of Machine Learning Research, vol. 206. PMLR, 2023, pp. 9417–9428.
>
> - S. Ren, X. Cao, Y. Wei, and J. Sun, “Global Refinement of Random Forest,” in Proceedings of the IEEE Conference on Computer Vision and Pattern Recognition (CVPR), 2015.
>
> - F. Alkhoury, S. Buschjäger, and P. Welke, “Splitting stump forests: tree ensemble compression for edge devices (extended version),” Machine Learning, vol. 114, no. 10, Aug. 2025, doi: 10.1007/s10994-025-06866-2.
>
> - BudgetPrune: F. Nan, J. Wang, and V. Saligrama, “Pruning Random Forests for Prediction on a Budget,” in Advances in Neural Information Processing Systems, D. Lee, M. Sugiyama, U. Luxburg, I. Guyon, and R. Garnett, Eds., Curran Associates, Inc., 2016.
>
> In addition, we identified three more papers that specifically target boosted tree ensembles. We consider these works directly relevant and have now included them in the related work section as well.
>
> - H. Wang, Z. Wu, X. Wang, L. Bian, and H. Jin, “HardGBM: A Framework for Accurate and Hardware-Efficient Gradient Boosting Machines,” IEEE Transactions on Computer-Aided Design of Integrated Circuits and Systems, vol. 42, no. 7, pp. 2122–2135, 2023, doi: 10.1109/TCAD.2022.3218509.
>
> - M. Alsharari, S. T. Mai, R. Woods, and C. Reaño, “Efficient Integer-Only-Inference of Gradient Boosting Decision Trees on Low-Power Devices,” IEEE Transactions on Circuits and Systems I: Regular Papers, vol. 72, no. 1, pp. 241–253, Jan. 2025, doi: 10.1109/tcsi.2024.3446582.
>
> - BitBoost: L. Devos, W. Meert, and J. Davis, “Fast Gradient Boosting Decision Trees with Bit-Level Data Structures,” in Machine Learning and Knowledge Discovery in Databases, U. Brefeld, E. Fromont, A. Hotho, A. Knobbe, M. Maathuis, and C. Robardet, Eds., Cham: Springer International Publishing, 2020, pp. 590–606. doi: 10.1007/978-3-030-46150-8_35.

---

> ### Author Response · Authors · 2025-11-21
> **Rebuttal - Part II**
>
> ### Further Baselines
>
> > W2: Continuation to above -- I think the absence of more baselines esp. in optimized tree implementations is a weakness.
>
> > Q2: Can you compare your work empirically to those papers?
>
> We focused our comparison on methods that are genuinely applicable to boosted decision trees. Most related approaches cannot be integrated meaningfully because they are designed either for random forests (e.g., Koschel et al., 2023; Buschjäger & Morik, 2023; Ren et al., 2015; Alkhoury et al., 2025) or for single decision trees (e.g., Bonsai, Kumar et al., 2017). Their implementations and frameworks (e.g., PyPrune) expect the output of a single tree to be a final prediction rather than a residual-correction step, making them incompatible with gradient boosting.
>
> Among the additional methods mentioned, Devos et al. focus primarily on inference speed. The two other works (Wang et al., 2023; Alsharari et al., 2025) do target compressing boosted trees, but both are tailored to FPGAs and rely on specialized quantization pipelines. Note that one of them does not provide any codebase, while the other requires proprietary AMD tools and hardware and is explicitly marked as “not fully ready to be implemented” (see https://github.com/malsharari/QATGBDT). Moreover, most of these methods cannot be trained under varying memory constraints, which makes a fair and consistent comparison with our extensive evaluation setup difficult.
>
> As outlined above, our method is intentionally designed to stay very close to standard XGBoost and LightGBM models. By adding only lightweight features and threshold penalizers and applying a simple storage scheme with shared thresholds and features, we preserve the original boosting architecture while gaining additional compression potential for memory-constrained devices. For the same reason, we use well-established implementations such as LightGBM as baselines, enabling a clear assessment of the compression benefits introduced by our approach without altering the underlying model class. We hope this clarifies our rationale and that our reasoning behind the chosen experimental setting adequately addresses the reviewer’s concern.

---

> > ### Comment · Reviewer_gDno · 2025-11-26
> >
> > I thank the authors for posting their response.
> >
> > From the response, I am satisfied with the additional literature survey.
> >
> > I understand and appreciate the authors response on challenges in empirically comparing with other methods.
> > However, I still hope if there is a way to empirically justify the strength of this paper compared to earlier work. I don't know if feasible or not -- if not gradient boosted trees -- how might the results (accuracy, time, memory) compare to RF pruning methods? If RF ones are better on all dimensions -- do we need gradient boosted trees?
> >
> >
> > I am also keeping a tab on the other reviewers points and your responses.

---

> > > ### Author Response · Authors · 2025-11-27
> > >
> > > Thank you very much for the fast response. We are happy to hear that we could meet your expectations regarding the literature extension.
> > >
> > > To address your concerns regarding further related works we have added additional tree-based baselines.
> > > In particular, we have trained RF models in a similar manner as the baseline GBDT methods. You can find additional experiments in the appendix of our manuscript (see Appendix D, page 16). As a pruning method we chose the one presented by Guo et al. (2018). We hope this addresses your remaining concerns sufficiently.
> > >
> > > H. Guo, H. Liu, R. Li, C. Wu, Y. Guo, and M. Xu, “Margin & diversity based ordering ensemble pruning,” Neurocomputing, vol. 275, pp. 237–246, Jan. 2018, doi: 10.1016/j.neucom.2017.06.052.

---

### Official Review · Reviewer_BpWP · 2025-10-31

**Soundness:** 3
**Presentation:** 3
**Contribution:** 2
**Rating:** 4
**Confidence:** 3

**Summary:**

This research introduces a novel compression scheme tailored for boosted decision tree ensembles to address the challenge of deploying machine learning models on compute-constrained IoT devices. The approach focuses on training compact models by strategically encouraging the reuse of features and thresholds. Experimental results demonstrate a substantial memory footprint reduction of 4–16x compared to standard LightGBM models without compromising performance. This optimization is crucial for enabling autonomous, low-power IoT applications such as remote monitoring, edge analytics, and real-time decision-making in isolated or energy-limited environments.

**Strengths:**

Penalizing the use of new features/thresholds encourages reuse across trees. The idea of feature reuse and threshold reuse is interesting and simple yet useful for sustainable ML. Introduction of a new loss function.

The work is very useful for doing efficient ML on resource-constrained devices.

Good sensitivity analysis.

I found the writing to be decent, and the paper is well structured.

The performance gains with minimal memory consumption compared to SOTA methods.

**Weaknesses:**

The part on memory layout based on encoding the information in a bit-wise manner is not novel.

No actual implementation on MCUs, which makes this a purely algorithmic work. A bit more analysis, including power consumption and energy efficiency, is required.

Domains requiring distinct rules (e.g., heterogeneous datasets) do not allow threshold reuse without performance loss.

The idea is good, but the utility of it is limited. For example, it does not make sense to train ML models on tiny MCU devices with 32 KB of RAM. The authors should motivate the real use case scenarios where their proposed method will be useful in the real world.

**Questions:**

What are some real-life scenarios where one will need to do training on MCUs?

---

> ### Author Response · Authors · 2025-11-21
> **Rebuttal - Part I**
>
> Thank you for your comments and questions. We are glad that you found our approach, experiments, and overall presentation useful. We provide detailed clarifications below and are happy to address any further questions.
>
> ### Novelty
>
> > W1: The part on memory layout based on encoding the information in a bit-wise manner is not novel.
>
> We agree that bit-wise node encoding is a known technique and do not claim novelty for this component. We describe it in detail because it is a key part of our overall memory layout, which, together with pointer-less storage, shared metadata, and lightweight feature/threshold penalizers, enables the substantial compression achieved by ToaD while keeping the models very close to standard XGBoost/LightGBM. This design allows us to preserve the original boosting architecture and evaluate compression gains fairly using well-established baselines such as LightGBM.
>
> To the best of our knowledge, no prior work combines bit-wise encoding with these additional mechanisms in a unified compression framework for boosted trees. If we have overlooked related work, we would welcome any pointers and will happily integrate them into the revised manuscript.
>
> ### Deployment Analysis
>
> > W2: No actual implementation on MCUs, which makes this a purely algorithmic work. A bit more analysis, including power consumption and energy efficiency, is required.
>
> Our work is primarily motivated by enabling boosted-tree models on highly resource-constrained devices, where memory (not latency or energy) is the primary deployment constraint. In many of our target applications, inference latency is not the limiting factor, whereas model size often determines whether deployment on the device is possible at all. Moreover, off-loading data typically incurs substantially higher energy costs than performing local inference (Muhoza et al., 2023) and adds transmission latency. By enabling stronger models to run locally, ToaD therefore implicitly reduces overall energy use.
>
> We fully agree that latency and energy measurements would further strengthen the empirical evaluation. However, a fair and representative comparison would require re-implementing all competing methods across multiple MCU platforms, an extensive engineering effort that is beyond the scope of the present study. We view this as an important direction for future work.
>
> Muhoza, A. C., Bergeret, E., Brdys, C., & Gary, F. (2023). Power consumption reduction for IoT devices thanks to Edge-AI: Application to human activity recognition. Internet of Things, 24, 100930. https://doi.org/10.1016/j.iot.2023.100930
>
> ### Performance Loss
>
> > W3: Domains requiring distinct rules (e.g., heterogeneous datasets) do not allow threshold reuse without performance loss.
>
> Threshold reuse does influence model performance, as also shown in our sensitivity analysis (see Figure 6). If the penalty term is chosen too aggressively, accuracy can indeed drop. This effect is naturally task- and dataset-dependent, which is why we recommend tuning the penalty value for each dataset.
>
> That said, our experiments indicate that threshold reuse is feasible for many practical settings. For real-valued features (for example, temperature), the exact numeric value of a threshold is often less important than the coarse partition it induces. This allows thresholds to be shared without a substantial loss in accuracy. For categorical features, dedicated thresholds are typically required. However, in such cases the number of possible thresholds is usually very small (often no more than a few dozen), which means the bit budget per threshold can still be kept low (e.g., around 10 bits).
>
> While performance can decline as threshold reuse increases, the resulting reduction in model size is substantial. Our experiments show that ToaD achieves a favorable trade-off between accuracy and memory footprint and ultimately outperforms alternative methods in this regard.

---

> ### Author Response · Authors · 2025-11-21
> **Rebuttal - Part II**
>
> ### Use Cases / Training on MCUs
>
> > W4: The idea is good, but the utility of it is limited. For example, it does not make sense to train ML models on tiny MCU devices with 32 KB of RAM. The authors should motivate the real use case scenarios where their proposed method will be useful in the real world.
>
> > Q1: What are some real-life scenarios where one will need to do training on MCUs?
>
> Thank you for raising this point. To clarify: we do not intend to perform training on MCUs. MCUs are used exclusively for inference, while training takes place on resource-rich hardware such as consumer PCs or high-performance computing clusters. We noticed that one sentence in the manuscript may have caused confusion. In line 118, we state that we “aim to minimize the memory footprint of the ensemble during the training process.” What we mean is that the final model is already optimized for memory during training, rather than being compressed afterwards through pruning or quantization. In other words, the training procedure directly produces a compact model suitable for MCU deployment, but the training itself is not performed on the MCU. We have clarified this aspect in the revised manuscript.
>
> Regarding real-world scenarios, our method applies to essentially any setting where decision-tree models are deployed on tiny devices, such as sensor-based applications in industrial monitoring, smart farming, environmental sensing, or predictive maintenance.

---

> > ### Comment · Reviewer_BpWP · 2025-11-25
> >
> > I have read the rebuttal, and my concern about the implementation of inference on the actual MCU is not addressed. I will raise my score if this work is implemented on an actual MCU and report the latency and power consumption.

---

> > > ### Author Response · Authors · 2025-11-27
> > >
> > > Thank you for your quick response. We will address your concerns about MCU deployment further. We are currently implementing the necessary methods and will evaluate the latency and energy of ToaD and competing methods. We will get back to you before the end of the rebuttal period.

---

> > > > ### Author Response · Authors · 2025-12-02
> > > >
> > > > We appreciate your feedback and the valuable perspectives you provided. The corresponding latency experiments have been added, and we direct you to the comprehensive response summary and revised manuscript for further information.

---

### Official Review · Reviewer_YmzG · 2025-10-31

**Soundness:** 3
**Presentation:** 2
**Contribution:** 4
**Rating:** 6
**Confidence:** 2

**Summary:**

This paper introduces Trees on a Diet (ToaD), a method for compressing GBDT for resource-constrained devices. It employs a regularization strategy that encourages the reuse of features and thresholds during training, coupled with a specialized, bit-wise memory layout. This layout uses global lookup tables and a pointer-less array representation to minimize storage. Evaluated on eight tabular datasets, ToaD compresses models that are 4–16× smaller than baseline LightGBM while maintaining comparable accuracy under tight memory constraints.

**Strengths:**

1. The work enables complex models like GBDTs to run on severely memory-constrained microcontrollers.
2. The proposed pointer-less array-based memory layout is highly suitable for microcontroller deployment, minimizing memory footprint and avoiding inefficient pointer chasing.
3. The method is orthogonal to many existing compressions like pruning and quantization, and easy to integrate into existing work
4. The experimental evaluation is comprehensive and sound

**Weaknesses:**

1. Potential inference latency overhead. The decoding process involving bit-level manipulations and lookups in global arrays may inherently more computationally expensive than the direct pointer-based traversal used in standard implementations. The paper would be significantly strengthened by an end-to-end latency evaluation.
2. Linear penalty is not motivated theoretically (e.g., from a Bayesian perspective) or empirically against other potential forms. Would a logarithmic penalty, which might better model diminishing costs, be more effective?
3. Several figures (e.g., Figure 6)  is so small that makes the text are hard to read. Furthermore, in Figure 5, the label is partially covered.

**Questions:**

1. The search over 32,076 configurations per dataset is computationally intensive. Do you have any insights or heuristics for navigating the hyperparameter more efficiently in practice?
2. For larger models that must reside in main memory (or SSD) and are evaluated on systems with hierarchical caches, could the non-sequential, lookup-heavy memory access pattern of your method lead to frequent cache misses or I/O bottlenecks compared to the more sequential access of standard array-based tree representations?

---

> ### Author Response · Authors · 2025-11-21
> **Rebuttal - Part I**
>
> Thank you for your positive feedback regarding contribution and evaluation of our work and especially for the constructive comments.
>
> ### Latency Overhead
>
> > W1: Potential inference latency overhead. The decoding process involving bit-level manipulations and lookups in global arrays may inherently be more computationally expensive than the direct pointer-based traversal used in standard implementations. The paper would be significantly strengthened by an end-to-end latency evaluation.
>
> Our work is motivated by scenarios in which the models must operate on extremely resource-limited devices, where memory usage determines whether deployment is possible at all. In these settings, model size is typically the dominant constraint, whereas inference latency is usually less critical. It is also worth noting that offloading data to external servers often incurs far higher energy costs than performing inference locally (Muhoza et al., 2023) and introduces additional communication latency. By enabling more expressive models to run directly on small microcontrollers, ToaD can therefore help reduce overall energy consumption in practical applications. From an implementation perspective, the added computational overhead is based on a few bit-level operations, and we expect the runtime and per-prediction energy costs to remain close to those of their standard counterpart.
>
> We agree that reporting latency and energy measurements would provide a more comprehensive empirical picture. Conducting such an evaluation in a fair and meaningful way, however, would require implementing all baseline methods across multiple MCU platforms, which constitutes a substantial engineering effort. While we view this as an important direction for follow-up work, it lies beyond the scope of the current study.
>
> We hope this clarifies our perspective and provides a transparent justification for the experimental focus of the paper.
>
> We have added a corresponding comment to our manuscript (see Footnote 2 in Section 3).
>
> Muhoza, A. C., Bergeret, E., Brdys, C., & Gary, F. (2023). Power consumption reduction for IoT devices thanks to Edge-AI: Application to human activity recognition. Internet of Things, 24, 100930. https://doi.org/10.1016/j.iot.2023.100930
>
> ### Penalty Function
>
> > W2: Linear penalty is not motivated theoretically (e.g., from a Bayesian perspective) or empirically against other potential forms. Would a logarithmic penalty, which might better model diminishing costs, be more effective?
>
> Thank you for raising this point. In fact, we experimented with other penalizers as well (see Footnote 3). Since the performance was very similar during our experimental evaluation, we decided to focus on the linear penalty term for the sake of simplicity.
>
> ### Figure Readability
>
> > W3: Several figures (e.g., Figure 6) is so small that makes the text are hard to read. Furthermore, in Figure 5, the label is partially covered.
>
> Thank you for pointing this out. We have increased the font sizes and have adapted the figures in the revised manuscript.
>
> ### Training Efficiency
>
> > Q1: The search over 32,076 configurations per dataset is computationally intensive. Do you have any insights or heuristics for navigating the hyperparameter more efficiently in practice?
>
>
> You are right that training 32,076 models per dataset is computationally very expensive. As explained in the paper, this large number comes from jointly sweeping over the maximum number of trees, maximum depth, and especially the two penalty terms. We used such a wide grid to provide a comprehensive and fair evaluation, but this is not required in practice.
> The reported results already allow us to narrow down the search space substantially. The multivariate sensitivity analyses (Figure 7 and Appendix D.2) show that threshold penalties between 4 and 4096 provide the best performance/memory trade-off. Restricting the search to this range alone reduces the search space by roughly a factor of six. Moreover, the almost square-shaped structure of the sensitivity plots indicates that choosing similar values for the feature penalty and the threshold penalty is generally sufficient. This means practitioners do not need to explore the full combination of both penalties; setting the feature penalty equal to the threshold penalty works well in most cases.
> In practice, strong configurations are typically obtained by
> - setting both penalty terms to the same value in the range 4 to 4096 (powers of 2), and
> - limiting the maximum tree depth to values between 2 and 64 (powers of 2).
>
> With these restrictions, one only needs to train 66 ToaD models instead of 32,076, while still having a very high chance of finding a competitive configuration. We will add these findings to the README of our code repository and in the next version of our manuscript.

---

> ### Author Response · Authors · 2025-11-21
> **Rebuttal - Part II**
>
> ### Applicability to Larger Models
>
> > Q2: For larger models that must reside in main memory (or SSD) and are evaluated on systems with hierarchical caches, could the non-sequential, lookup-heavy memory access pattern of your method lead to frequent cache misses or I/O bottlenecks compared to the more sequential access of standard array-based tree representations?
>
> Thank you for this interesting question. Yes, this could happen. The most obvious strategy to bypass this is to store the lookup-tables in the RAM and load trees individually from main memory/SSD to process them one after another. With exploding lookup-tables this is not feasible and would indeed result in heavy lookup memory access patterns. However, our approach is intended to prevent the need for storing a model in the main memory as we aim to load models entirely to RAM.

---

> > ### Comment · Reviewer_YmzG · 2025-11-26
> > **Response by Reviewer**
> >
> > Thank the authors for their response. While I understand the focus on memory, I believe the paper's contribution would be substantially strengthened by including latency analysis. Given that the core motivation involves deployment on resource-constrained edge/mobile devices, latency is a co-equal metric with memory in evaluating such trade-offs. Incorporating even a basic latency benchmark on the one MCU platform would greatly solidify the paper's practical claims and relevance to the field.

---

> > > ### Author Response · Authors · 2025-11-27
> > >
> > > We appreciate the prompt feedback and plan to investigate the latency of our approach further. Specifically, we are developing and evaluating methods to assess ToaD's latency and energy efficiency compared to existing methods. We will report on our findings by the end of the rebuttal period.

---

> > > > ### Author Response · Authors · 2025-12-02
> > > >
> > > > Thank you again for your constructive feedback and helpful observations. We have included the relevant latency experiments and refer to the overall response summary as well as the updated manuscript for additional details.

---

### Official Review · Reviewer_GwkK · 2025-11-01

**Soundness:** 2
**Presentation:** 3
**Contribution:** 2
**Rating:** 4
**Confidence:** 4

**Summary:**

In this article, the authors propose Trees on a Diet (ToaD), a training‑time compression framework for Gradient-Boosted Decision Tree (GBDT) ensembles, targeting microcontrollers and, more generally, resource-constrained edge/embedded devices.
The core ideas are: *i)* adding linear regularizers that penalize the introduction of new features and new thresholds across the ensemble to encourage reuse during tree growth, and *ii)* deploying a pointer‑less bit‑wise memory layout with global lookup tables for feature thresholds and global leaf values shared across all trees.
The results show that ToaD matches the competitor's performance, reporting 4 to 16 times lower memory usage for the same accuracy in the most relevant memory range (≤128 KB).

**Strengths:**

The work tackles a concrete real-world problem: DTs on microcontrollers, where RAM and flash budgets are very limited.

The article is well-written, structured, and easy to follow.

The proposed method is described in sufficient detail.
Specifically, the introduction of ensemble-level penalties using features and thresholds is a simple yet effective idea to induce parameter reuse, which complements post-training pruning/quantization.
Furthermore, the design choice to store the threshold bit-width and numeric type per feature provides a flexible precision/size trade-off.

**Weaknesses:**

The primary motivation for this work is the deployment of GBDT on resource-constrained devices, where memory is a critical constraint, as well as latency (and energy consumption).
The authors provide results of memory savings, but do not present any experimental results on inference speed (and energy consumption).
Without this analysis, the practical utility of ToaD for real-time edge applications remains unproven.

The authors state that the $RF$ is “the ratio between the global number of values and the sum of the nodes and leaves” (line 371).
Thus, if values are reused effectively, the number of global values becomes smaller, while the number of nodes and leaves remains fixed; therefore, a good reuse would produce $RF<1$.
At line 374, the interpretation in the text says the opposite, implying that $RF$ should be (#nodes + #leaves)/(#global values), *i.e.*, the inverse of the statement above.

Memory for baselines is computed under a simplified node model that includes two child pointers, whereas ToaD benefits from a pointer‑less encoding and global sharing.
This risks giving ToaD an advantage in the comparison.

Experiments use a single 80/20 split per dataset with large sweeps and report the best points within memory limits; however, no statistical significance values are presented.

The authors cite several other relevant works on tree compression and optimization (*e.g.*, Koschel et al., 2023, and Buschjäger & Morik, 2023) in their related work section, but do not include them in the experiments.

**Questions:**

The authors should:
1) Provide inference latency and energy consumption (per prediction) on representative MCUs (*e.g.*, the mentioned ARM Cortex‑M4 @ 48 MHz) for ToaD versus baselines.
2) Clarify the reuse factor formula to align with the intended interpretation.
3) Report memory results under a unified layout, or at least discuss the potential advantages of ToaD against baseline models.
4) Report mean±std for accuracy at each memory budget, and compare against other relevant presented works only mentioned in the state-of-the-art.

---

> ### Author Response · Authors · 2025-11-21
> **Rebuttal - Part I**
>
> We would like to thank the reviewer for the comments and suggestions. We have revised the paper and tried to address all comments and questions. Our detailed responses to the reviewer’s comments are provided below.
> ### Inference Latency and Energy Consumption
> > Q1: Provide inference latency and energy consumption (per prediction) on representative MCUs (e.g., the mentioned ARM Cortex‑M4 @ 48 MHz) for ToaD versus baselines.
>
> > W1: The primary motivation for this work is the deployment of GBDT on resource-constrained devices, where memory is a critical constraint, as well as latency (and energy consumption). The authors provide results of memory savings, but do not present any experimental results on inference speed (and energy consumption). Without this analysis, the practical utility of ToaD for real-time edge applications remains unproven.
>
> Our work is primarily motivated by enabling boosted-tree models on highly resource-constrained devices, scenarios in which memory, rather than latency or energy, constitutes the dominant bottleneck. In many of our target applications, inference latency is not the limiting factor, whereas model size directly determines whether a model can be deployed on the device at all. It is also important to note that sending data off-device typically incurs energy costs that significantly exceed those of local inference (Muhoza et al., 2023) and introduces transmission latency. By allowing stronger models to run locally on tiny hardware, ToaD implicitly contributes to reducing overall energy demand in such settings. From a technical perspective, our compression scheme introduces only minimal computational overhead (e.g., a small number of bit-wise operations, typically on the order of single CPU cycles). We therefore expect the corresponding impact on runtime and energy per prediction to be minor.
>
> Muhoza, A. C., Bergeret, E., Brdys, C., & Gary, F. (2023). Power consumption reduction for IoT devices thanks to Edge-AI: Application to human activity recognition. Internet of Things, 24, 100930. https://doi.org/10.1016/j.iot.2023.100930
>
> We fully agree that reporting latency and energy measurements would further strengthen the empirical evaluation. A comprehensive and fair latency/energy comparison across baselines would, however, require implementing all competing methods on multiple MCU architectures to ensure representative measurements. While we consider such an evaluation relevant and plan to explore it in future work, we believe that conducting this extensive cross-platform engineering effort is beyond the scope of the present study.
>
> We have added a corresponding comment to our manuscript (see Footnote 2 in Section 3).
>
> ### Reuse Factor
>
> > Q2: Clarify the reuse factor formula to align with the intended interpretation.
>
> Thank you very much for pointing this out. You are absolutely right, there was a mistake in the text. We have revised the manuscript accordingly so that the text and the formula are now consistent (see the beginning of Section 4.3 in the updated version of our manuscript).

---

> ### Author Response · Authors · 2025-11-21
> **Rebuttal - Part II**
>
> ### Memory Layout
>
> > W3: Memory for baselines is computed under a simplified node model that includes two child pointers, whereas ToaD benefits from a pointer‑less encoding and global sharing. This risks giving ToaD an advantage in the comparison.
>
> > Q3: Report memory results under a unified layout, or at least discuss the potential advantages of ToaD against baseline models.
>
> We acknowledge that different node layouts can influence the apparent memory footprint of tree-based models. Our goal, however, is not to change the model class but to minimize the per-node storage cost of boosted trees, a bottleneck inherent to all GBDT models. In this sense, ToaD should be viewed as a set of orthogonal memory-reduction techniques that can, in principle, also be applied to baseline models (which might not foster feature/threshold reuse though).
>
> In a naïve layout, each node typically stores two 32-bit child pointers, a 32-bit feature index, and a 32-bit threshold (around 128 bits per node). ToaD replaces this design with (i) pointer-less tree storage, (ii) compressed feature indices, and (iii) quantized thresholds/leaf values, plus some globally shared metadata. Under ideal conditions, this reduces storage to only a few bits per node, leading to compression factors on the order of 40-50x (e.g., in case only 1-2 bits are needed to store the feature index/threshold per node of a tree). Note that the particular compression factor is highly-dataset dependent. For instance, given the wine multiclass dataset, ToaD achieves a compression factor of about 16x for various memory limits.
>
> Between these two extremes (baseline naïve vs. ToaD-optimal) exists a continuum of intermediate layouts, such as pointer-less trees with float16 thresholds, or reduced-bit feature indices (e.g., only 10 bits are needed for datasets with at most 256 features). Some of these optimizations would benefit certain baselines as well, but their effectiveness depends strongly on tree shape. A pointer-less layout, for example, is advantageous only when trees are near-complete, a property typical for shallow boosted trees, but not guaranteed for all methods. Likewise, ToaD occasionally wastes memory in unused blocks if a trained tree is not perfectly complete.
>
> To address your point, we have added the memory consumption of LightGBM in an array-based memory layout in our comparison and refer to the updated Figure 4 of our manuscript. As you can see, “LightGBM array-based” is not always advantageous over its pointer-based variant LightGBM.
>
> ### Random Data Splits & Repetitions
>
> > Q4: Report mean±std for accuracy at each memory budget ....
>
> Thank you for pointing this out. We have begun repeating our experiments across 10 independent train/test splits. This process is highly time-consuming and currently constrained by available computational resources. We will share preliminary results as soon as they are ready, and we expect to do so by the beginning of next week.
>
> ### Additional Comparisons
>
> > W5: The authors cite several other relevant works on tree compression and optimization (e.g., Koschel et al., 2023, and Buschjäger & Morik, 2023) in their related work section, but do not include them in the experiments.
> > Q4: [...] compare against other relevant presented works only mentioned in the state-of-the-art.
>
> We acknowledge your point. As pointed out above, the main goal of our work is to not change the model class, but to minimize the memory footprint of classical boosted decision trees. For these reasons, we have compared ToaD with the closest competitors, namely baseline LightGBM, quantized LightGBM, cost-efficient gradient boosting, and minimal cost-complexity pruning. We believe that this comparison is meaningful, since it allows a direct comparison of the well-known LightGM library, and the potential memory savings obtained through ToaD.
>
> We would also like to refer to our response to Reviewer gDno (“Further Baselines”) for additional details and clarification.

---

> > ### Comment · Reviewer_GwkK · 2025-11-27
> >
> > Dear authors, in general, thank you for the clarification and the considerable work that clearly went into the rebuttal.
> >
> > That said, I am not entirely convinced by the idea that latency can be treated as a "secondary" measurement in IoT and other resource-constrained scenarios.
> > Even when memory is the main bottleneck, latency often remains a critical metric for many real-world deployments.
> > For this reason, I still believe that including empirical evidence on latency would substantially strengthen the evaluation of the work and better support the claims about the practical use.
> > However, I appreciate your openness to future latency and energy measurements.
> >
> > Regarding the additional experiments (mean ± std across multiple splits), I really appreciate the effort involved and look forward to seeing the results as soon as they are available, as indicated in your response.

---

> > > ### Author Response · Authors · 2025-11-27
> > >
> > > Thank you for the fast answer. As we already stated we are running experiments on further data splits. As our detailed evaluations require many training runs we did not finish the training for as many data splits as planned. So far we can report on 5 different splits per datasets, except for covtype multiclass where just 4 runs are finished. As you can see in our update to Figure 4 (page 7), the results are mostly stable over multiple runs. We can observe a larger standard deviation for smaller datasets but the overall pattern stays the same with ToaD outperforming the baseline methods.
> > >
> > > We will try to further address your concerns regarding latency. We are currently implementing respective methods and will try to evaluate latency and energy of ToaD and competing methods. We will get back to you until the end of the rebuttal period.

---

> > > > ### Author Response · Authors · 2025-12-02
> > > >
> > > > Thank you once again for your thoughtful feedback and valuable insights. We have incorporated the corresponding latency experiments and kindly refer you to the overall response summary and the revised manuscript for further details.

---

### Author Response · Authors · 2025-11-21
**Author Response Summary**

We thank all reviewers for their thorough evaluation and the many insightful comments and suggestions. We have carefully revised the paper accordingly and highlight the main changes below. All manuscript modifications are marked in blue in the updated version.

**1. Clarifications and Corrections**
  - **Scope (memory vs. latency/energy)**. Clarified that the work targets MCU inference under strict memory budgets. Training is always done off-device. Added a short explanation of why latency/energy measurements are not included and what a fair evaluation would require.
  - **Reuse factor**. Corrected and streamlined the definition so that text and formula are consistent.
  - **Memory layout discussion**. Clarified how ToaD’s layout relates to baseline layouts and when pointer-less storage or reduced-bit indices could also benefit baselines.
  - **Practical configuration guidance**. Added brief heuristics for choosing penalty strengths and depth ranges based on the sensitivity analyses.

**2. Related Work and Baselines**
- **Expanded related work**. Added additional methods on compressed tree ensembles and hardware-efficient GBDTs to better situate the contribution.
- **Baseline selection**. Added a clear explanation of why several recent methods could not be included experimentally (e.g., unavailable or incomplete code, proprietary toolchains, incompatible with boosted-tree training).

**3. Figures and Readability**
- **Improved formatting**. Increased font sizes, fixed label overlaps, and updated appendix figures for clearer readability.

**4. Additional Experiments**
- **Figure 4 updated**. Added LightGBM (array-based) to separate the effect of the memory layout from the penalizers (see Page 7).
- We have begun rerunning all experiments across 10 independent train/test splits, each with a full hyperparameter search. This process is very time-consuming and currently constrained by available computational resources. We will share preliminary results as soon as they are available and expect to do so by the beginning of next week.

Thank you again for your careful reading and valuable comments. We are happy to clarify any remaining points.

---

### Author Response · Authors · 2025-12-02
**Discussion Summary**

As the review discussion phase concludes, we would like to summarize the concerns raised and our corresponding rebuttals, complementing the previously submitted ‘Author Response Summary'. In addition to the extensions mentioned there, we have included further experiments, explained in the following.

1. **Experiments on different data splits** have shown that our results generalize over varying data contributions with higher standard deviation for smaller datasets but very stable results for larger datasets.
2. **Comparison to (pruned) random forests** classification tasks reveal ToaD (and boosted trees in general) mostly outperform both, standard and pruned random forests, especially for binary classification.
3. First prototypical results on the **latency of our approach** where we deployed ToaD on microcontrollers. The inference time was measured and compared to the latency of a baseline LightGBM model.  Preliminary prototype implementations indicate that as expected the inference latency increases (e.g., ~0.1 ms for ToaD compared to ~0.5 ms for the baseline LightGBM model on the Arduino Nano 33 BLE; see result table below). However, in typical real-world deployment scenarios, where the device must wake from sleep mode, perform sensor measurements, run inference, transmit data (noting that establishing a connection to a network such as LoRaWAN can already take several seconds), and then return to sleep mode, the contribution of the inference step to overall latency is **negligible**. In our experiments, for instance, generating input values randomly already required more time than the inference itself. When using actual sensors, data acquisition takes substantially longer, making the relative share of inference latency even smaller.
We added the preliminary results of the experiments to the appendix of our manuscript and would like to refer to the respective code in the provided [anonymous repository](https://anonymous.4open.science/r/ToaD/experiments/latency/README.md).

| Hardware            | ToaD Inference Time | LightGBM Inference Time|
|---------------------|------------|----------|
| XIAO ESP32S3        | 137.1 $\micro$s       | 17.6 $\micro$s   |
| Arduino Nano 33 BLE | 512.9 $\textmu{}$s     | 102.2 $\micro$s  |

We thank the reviewers again very much for their constructive feedback that helped to strengthen our work. With the added experiments and further extensions to the manuscript we think we addressed all of the reviewers' concerns. All changes to the updated manuscript are highlighted with blue text. We are currently conducting additional, more detailed experiments and will be happy to incorporate the updated results into the final manuscript upon acceptance.

---

### Meta-Review · Area_Chair_KALX · 2026-01-13

**Summary:**

**Summary**

The paper presents Trees on a Diet (ToaD), a compression framework for Gradient-Boosted Decision Trees (GBDT) aimed at resource-constrained devices. It introduces linear regularizers to promote feature and threshold reuse during training and employs a pointer-less bit-wise memory layout with global lookup tables, achieving 4 to 16 times lower memory usage while maintaining accuracy. The method is evaluated on eight datasets, demonstrating significant model size reduction compared to LightGBM, making it suitable for IoT applications that require efficient memory usage without sacrificing performance. Overall, ToaD effectively addresses the challenges of deploying machine learning models in low-power environments.

**Strengths**
- **Practical Relevance and Innovation**: It addresses a significant real-world challenge of deploying decision trees on microcontrollers with limited RAM and flash memory. The introduction of ensemble-level penalties for feature and threshold reuse is a novel approach that enhances parameter efficiency and complements existing techniques like pruning and quantization.

- **Clear Presentation and Comprehensive Evaluation**: The article is well-structured and easy to follow, with a detailed description of the proposed method. The experimental evaluation is thorough, demonstrating the effectiveness of the approach with rigorous results and sensitivity analysis.

- **Memory Efficiency and Integration**: The proposed pointer-less array-based memory layout is specifically designed for microcontroller deployment, minimizing memory usage while maintaining performance. The method's compatibility with existing compression techniques makes it a versatile solution for efficient machine learning on resource-constrained devices.

**Weaknesses**
- **Lack of Inference Speed and Energy Consumption Analysis**: The authors provide memory savings results but do not include experimental data on inference speed or energy consumption, leaving the practical utility of ToaD unproven for real-time applications.

- **Potential Misinterpretation of Metrics**: There is a contradiction in the interpretation of the ratio of global values to nodes and leaves, which could lead to confusion regarding the effectiveness of value reuse.

- **Simplified Baseline Model**: The memory computation for baselines uses a simplified node model that may unfairly advantage ToaD due to its pointer-less encoding and global sharing.

- **Insufficient Experimental Rigor**: The experiments rely on a single dataset split without presenting statistical significance, and relevant works on tree compression are not included in the experiments, limiting the robustness of the findings.


**Decision**

The paper is recommended for acceptance as it tackles a critical real-world challenge: deploying decision trees on microcontrollers with limited RAM and flash memory. Additionally, it significantly reduces memory usage while maintaining a certain level of accuracy. To enhance the paper further, it would be beneficial to include the computational cost of each feature, as this would expedite the overall process.

**Reviewer Concerns:**

The author's feedback effectively addressed most concerns, enhancing the overall quality of the paper. Additionally, they clarified several ambiguous points, leading most reviewers to consider a slight increase in their scores.

**Reviewer Scores:**

The author's feedback effectively addressed most concerns, enhancing the overall quality of the paper. Additionally, they clarified several ambiguous points, leading most reviewers to consider a slight increase in their scores.

---

### Decision · Program_Chairs · 2026-01-26

Accept (Poster)